# Polyfunctionality and breadth of HIV-1 antibodies are associated with delayed disease progression

**Marloes Grobben**[1,2], **Margreet Bakker**[1,2], **Angela I. Schriek**[1,2], **Liesbeth J.J. Levels**[1,2], **Jeffrey C. Umotoy**[1,2], **Khadija Tejjani**[1,2], **Mariëlle J. van Breemen**[1,2], **Ryan N. Lin**[3], **Steven W. de Taeye**[1,2], **Gabriel Ozorowski**[3], **Neeltje A. Kootstra**[2,4], **Andrew B. Ward**[3], **Stephen J. Kent**[5,6], **P. Mark Hogarth**[7,8], **Bruce D. Wines**[7,8], **Rogier W. Sanders**[1,2,9], **Amy W. Chung**[5]\*, **Marit J. van Gils**[1,2]\*

1 Amsterdam UMC, location University of Amsterdam, Department of Medical Microbiology and Infection Prevention, Amsterdam, The Netherlands, 2 Amsterdam Institute for Immunology and Infectious Diseases, Amsterdam, The Netherlands, 3 The Scripps Research Institute, Department of Structural Biology and Computational Biology, La Jolla, California, United States of America, 4 Amsterdam UMC, location University of Amsterdam, Department of Experimental Immunology, Amsterdam, The Netherlands, 5 The Peter Doherty Institute for Infection and Immunity, The University of Melbourne, Department of Microbiology and Immunology, Melbourne, Australia, 6 Alfred Hospital and Central Clinical School, Monash University, Melbourne Sexual Health Centre and Department of Infectious Diseases, Melbourne, Australia, 7 Burnet Institute, Immune Therapies Group, Melbourne, Australia, 8 Central Clinical School, Monash University, Department of Immunology, Melbourne, Australia, 9 Weill Medical College of Cornell University, Department of Microbiology and Immunology, New York, New York, United States of America

\* awchung@unimelb.edu.au (AWC); m.j.vangils@amsterdamumc.nl (MJG)

**Data Availability Statement:** All data deposited in an open access public database at https://doi.org/10.5281/zenodo.14108392. All other relevant data

## Abstract

HIV-1 infection leads to chronic disease requiring life-long treatment and therefore alternative therapeutics, a cure and/or a protective vaccine are needed. Antibody-mediated effector functions could have a role in the fight against HIV-1. However, the properties underlying the potential beneficial effects of antibodies during HIV-1 infection are poorly understood. To identify a specific profile of antibody features associated with delayed disease progression, we studied antibody polyfunctionality during untreated HIV-1 infection in the well-documented Amsterdam Cohort Studies. Serum samples were analyzed from untreated individuals with HIV-1 at approximately 6 months (n = 166) and 3 years (n = 382) post-seroconversion (post-SC). A Luminex antibody Fc array was used to profile 15 different Fc features for serum antibodies against 20 different HIV-1 envelope glycoprotein antigens and the resulting data was also compared with data on neutralization breadth. We found that high HIV-1 specific IgG1 levels and low IgG2 and IgG4 levels at 3 years post-SC were associated with delayed disease progression. Moreover, delayed disease progression was associated with a broad and polyfunctional antibody response. Specifically, the capacity to interact with all Fc γ receptors (FcγRs) and C1q, and in particular with FcγRIIa, correlated positively with delayed disease progression. There were strong correlations between antibody Fc features and neutralization breadth and several antibody features that were associated with delayed disease progression were also associated with the development of broad and potent antibody neutralization. In summary, we identified a strong association between broad, polyfunctional antibodies and delayed disease progression. These findings

are within the manuscript and its Supporting Information files.

**Funding:** This research was supported by a Work Visit Grant of the Amsterdam institute for Infection and Immunity (to M.G.), by a National Health and Medical Research Council (NHMRC) Investigator grant (to A.W.C), by Amsterdam UMC through the AMC Fellowship (to M.J.v.G.) and by the Netherlands Organization for Scientific Research (NWO) through an Aspasia grant (to M.J.v.G.). Electron microscopy work was supported by the Bill and Melinda Gates Foundation and CAVD network INV-002916 (to A.B.W.). The funders had no role in study design, data collection and analysis, decision to publish, or preparation of the manuscript.

**Competing interests:** The authors have declared that no competing interests exist.

contribute new information for the fight against HIV-1, especially for new antibody-based therapy and cure strategies.

## Author summary

Despite the availability of effective treatment, HIV-1 still causes significant mortality and morbidity across the globe. Alternative ways for protection or treatment against HIV-1 are needed. Antibody-mediated effector functions could potentially contribute to the effectiveness of novel approaches. However, we need more information on which antibody properties are associated with these potential beneficial functions of antibodies. Studying antibody responses during natural HIV-1 infection can provide guidance on this topic. This study identified several antibody properties that were associated with reduced HIV-1 disease progression in a large cohort of individuals with untreated HIV-1. High levels of IgG1 and low levels of IgG2 and IgG4, broad and polyfunctional antibodies and interaction with immune proteins FcyRs and C1q, were all associated with delayed disease progression. Subsequently, effective strategies against HIV-1 will likely require multiple different components, and the antibody properties described in this study may contribute to a more detailed bio-molecular roadmap for antibody-based strategies for HIV-1 prevention, therapy and cure.

## Introduction

Despite the availability of effective treatment, the HIV-1 epidemic is still ongoing and the need for a vaccine, cure and/or alternative treatments is evident. This remains a monumental challenge: over 40 years of intensive research has not yet led to clinical success in either the vaccine or cure field [1]. Studying immune responses during the natural course of HIV-1 infection can lead to new insights, particularly in the rare individuals that do not develop acquired immunodeficiency syndrome (AIDS) in the absence of treatment. Cohort studies have shown that the development of broadly neutralizing antibodies (bNAbs) after HIV-1 infection is not associated with delayed disease progression [2–4]. However, bNAbs can protect non-human primates and humans from HIV-1 acquisition [5–7]. Additional studies have revealed that the development of antibody-mediated effector functions correlate with HIV-1 control [8–13]. However, vaccine efficacy studies focused on antibody-mediated effector functions also have not yet led to clinical success.

Antibody-mediated effector functions are induced when the Fc region of antigen-opsonized antibodies interact with Fc γ receptors (FcγRs) on immune cells. This interaction can lead to a plethora of biological responses depending on the cell type on which the receptor(s) reside. Functions can include antibody dependent cellular phagocytosis (ADCP), antibody-dependent cellular cytotoxicity (ADCC), cytokine and chemokine release, and modulation of B and T cell responses [14]. In addition, antibodies can also interact with C1q protein and activate the complement cascade. This can lead to opsonization of target antigen, recruitment of immune cells and lysis of cell or viral membranes via pore formation (antibody-dependent complement-dependent cytotoxicity; ADCDC).

Antibody-mediated effector functions are regulated by balanced interaction with activating and inhibitory FcγRs [15]. FcγRI is the only high affinity receptor and can activate dendritic cells and phagocytes [16]. FcγRIIa is the most widely expressed FcγR, it plays a dominant role

in induction of ADCP activity, but can also mediate ADCC activity [17,18]. FcγRIIIa is present on neutrophils, NK cells, monocytes and macrophages. Interaction with FcγRIIIa is most closely related to ADCC activity, but it can also contribute to ADCP [19,20]. Both FcγRIIa and FcγRIIIa are represented in the human population as polymorphic variants of high (FcγRIIa-H131 and FcγRIIIa-V158) and low affinity (FcγRIIa-R131 and FcγRIIIa-F158) for IgG, with up to 50/50 prevalence in some populations [21]. FcγRIIb is expressed on B cells and most phagocytes and it is the only inhibitory FcγR. FcγRIIb can also be involved in antigen presentation [22]. FcγRIIIb is present on neutrophils and can trigger their activation [23].

An important additional benefit of effector function-mediating antibodies is that they can both mediate clearance of virus as well as lysis of HIV-1 infected cells. The effectiveness of antibody effector functions is determined by a range of factors. Antigen specificity (including affinity, valency and density of the targeted epitope) is important, but the effectiveness of these functions is also determined by the capacity of the antibody Fc tail to engage the different FcγRs, which is influenced by antibody isotype and subclass as well as antibody Fc polymorphisms/allotypes, Fc region glycosylation, orientation of binding to the antigen, as well as allosteric interactions in immune complexes formed in polyclonal serum [18,24,25]. The profile of FcγR binding is unique to each IgG subclass [26]. In blood, IgG1 is most prevalent, followed by IgG2, while IgG3 and IgG4 only comprise a small compartment [27]. However, IgG3 followed by IgG1, are the most potent mediators of antiviral activity through antibody-mediated effector functions while IgG2 and IgG4 are known to be weaker mediators of these functions [28]. Therefore, the frequency and proportion of antigen-specific antibodies of different subclasses also determines the functionality of a polyclonal antibody response. The duration of an infection is an important factor in the distribution of subclasses: class switching occurs in the order IgG3, IgG1, IgG2, IgG4 and therefore the antibody repertoire is destined to move towards less functional subclasses [29].

Effector functions can be mediated by both neutralizing and non-neutralizing antibodies (NAbs and non-NAbs) [30–32]. During the course of HIV-1 infection, non-NAbs are very abundant and appear quickly after infection [33], while bNAbs only develop in 10–30% of untreated HIV-1 infected individuals after 2–3 years [34]. Antibody-mediated effector functions can alter the course of HIV-1 infection: non-neutralizing antibodies can induce immune escape and reduce the number of transmitter/founder viruses [35–37]. Antibody-mediated effector functions can also mediate protection against Influenza, SARS-CoV-2, Marburg and Lassa virus [38–42]. When passively administered, bNAbs can protect from infection or delay viral rebound after treatment interruption [6,7,43]. However, abrogation of the supplemental antibody-mediated effector functions from passively transferred bNAbs can reduce both bNAb-mediated protection against HIV-1 infection and post-infection HIV-1 control by bNAbs [31,32,44].

However, in a polyclonal antibody response, the effect of different antibody functions is more complex. Positive correlations of serum antibody-mediated effector functions with delayed disease progression have been observed, but often with limited patient numbers, uncertainty about the time of sampling after infection and/or no information on long-term disease progression. Strong correlations between antibody-mediated effector functions and current CD4+ T cell counts and/or viral load were demonstrated in small patient groups without a known time of infection [9,10,45,46]. Some studies did not find direct associations of any single effector function but found that Fc polyfunctionality was associated with delayed disease progression [47,48]. Thus, there is a need for a more comprehensive study on the antibody response in a well-characterized cohort of untreated individuals with HIV-1 to further delineate the antibody profile that shows these associations.

Detailed information on antibody Fc properties in people living with HIV-1 is still lacking, in particular in relation to clinical information including time of infection and time to disease progression. Moreover, more information is needed on the possibility of eliciting antibodies that both exert effector functions and have neutralizing capacity. Hence, we studied the induction of polyfunctional antibody responses during untreated HIV-1 infection in 382 individuals of the well-documented Amsterdam Cohort Studies. Using 20 different HIV-1 antigens, we assessed the association between 15 different antibody Fc properties and disease progression as well as the coordination between these antibody Fc properties and with neutralization capacity in polyclonal serum. In this study, we found that high levels of IgG1 antibodies and low levels of IgG2 and IgG4 are associated with delayed disease progression. These associations were present for antibody interactions with all FcγRs and C1q protein, but especially for FcγRIIa. Moreover, we observed that Fc polyfunctionality and breadth are associated with delayed disease progression and that Fc polyfunctionality also coincides with higher neutralizing antibody potency and breadth. The outcomes of this study present an antibody profile that provides a more detailed bio-molecular roadmap for new antibody-based strategies for HIV-1 prevention, therapy and cure.

## Results

### High frequency and levels of envelope glycoprotein-specific antibody responses with various features detected 3 years after seroconversion

We analyzed sera from participants of the Amsterdam Cohort Studies with untreated HIV-1 to investigate the relationship between antibody functionality and HIV-1 disease progression. Sera were available at approximately 3 years post-seroconversion (SC) (36 months, range 21–56 months) from 382 individuals. Serum from 166 of these individuals was also available at approximately 6 months post-SC (range 3–11 months). The clinical characteristics of this subset were comparable to the full cohort (Table 1). Individuals were divided in 4 groups based on the time between SC and AIDS diagnosis, and demographics were comparable between the groups (S1 Table).

To allow characterization of HIV-1 specific antibody responses, we produced and purified monomeric gp120 and trimeric stabilized SOSIP gp140 from subtype B strain 92BR020. Blue native PAGE gels confirmed that the gp120 and the SOSIP trimer had the expected sizes (S1A Fig). Negative-stain electron microscopy (NS-EM) confirmed that the SOSIP protein formed trimers (S1B Fig). 92BR020 SOSIP and gp120 conjugated to Magplex beads showed distinct and expected antigenic profiles when exposed to several monoclonal antibodies (mAbs), with binding of quaternary dependent mAbs PGT145 and PGT151 to the SOSIP trimer and higher binding of the V3 targeting mAbs to the gp120 protein (S1C Fig). Using a Luminex assay [49],

**Table 1. The total number of participants, data and availability of data related to disease progression.** Abbreviations: n: number; NA: not applicable; AIDS: acquired immunodeficiency syndrome; cp/mL = viral copies per milliliter.

| | Sample at 6 months post-seroconversion available | | | Sample at 3 years post-seroconversion available | | |
|---|---|---|---|---|---|---|
| | n | median | range | n | median | range |
| Samples | 166 | NA | NA | 382 | NA | NA |
| AIDS diagnosis known | 102 | NA | NA | 268 | NA | NA |
| Undiagnosed but followed-up without AIDS for >11 years | 13 | NA | NA | 44 | NA | NA |
| Age at seroconversion (years) | 166 | 35 | 20–57 | 382 | 34 | 19–57 |
| Setpoint viral load (cp/mL) | 166 | 20000 | 50–920,000 | 370 | 19373 | 50–1,300,000 |
| Setpoint CD4+ T cell count (cells/μl) | 164 | 500 | 10–1,820 | 351 | 510 | 10–1,820 |

we analyzed all serum samples for HIV-1 specific antibody types IgG, IgA, IgM, IgG1-4 and HIV-1 specific antibody capability to interact with FcγRs (FcγRI, FcγRIIa, FcγRIIb, FcγRIIIa, FcγRIIIb) and C1q protein. Nearly all 92BR020 gp120- and SOSIP-specific IgG subtype levels increased between 6 months and 3 years post-SC, with an exception for gp120-specific IgG3, which decreased (Fig 1A). Interaction of serum 92BR020 gp120- and SOSIP-specific antibodies with FcγRs and C1q protein, indicative of the ability to induce ADCP, ADCC and ADCDC, were also all increased between 6 months and 3 years post-SC (Figs 1A and S2A). Responder rates were high for 92BR020 gp120 at 3 years post-SC (median: 97%), intermediate for 92BR020 SOSIP at 3 years post-SC (median: 86%) and 92BR020 gp120 at 6 months post-SC (median: 80%) and lowest for 92BR020 SOSIP at 6 months post-SC (median: 53%) (S2B Fig).

## Envelope glycoprotein-specific FcγR engagement and IgG1 antibodies correlate with delayed disease progression

Antibody subclass distribution and antibody-mediated effector functions were previously associated with protection against disease progression [9,47]. Therefore, we investigated the association between disease progression in our cohort and Fc features of the antibody response. At 3 years post-SC, antigen-specific IgG1 and IgG3 levels and the ability of serum antibodies to interact with FcγRs and C1q was significantly higher for individuals with a longer time to AIDS (data included in the online repository). No significant differences in antibody features were found at 6 months post-SC but there were some interesting trends. For example, there was a trend for lower IgG and IgG1 levels at 6 months post-SC in individuals with a longer time to AIDS. For the full cohort at 3 years post-SC, we found significant spearman correlations of time between SC and AIDS with 92BR020 gp120-specific Fc features for IgG1, FcγRI, FcγRIIa, FcγRIIIa, FcγRIIb and C1q, with correlation coefficients between 0.18 and 0.28 (Fig 1B). Correlations with SOSIP-specific features were slightly weaker. Low viral load and high CD4+ T cell count at setpoint are known to be related to slower disease progression and correlations between these markers and low levels of IgG2 and IgG4 were identified.

Additionally, we found that IgG2 and IgG4 poorly correlated with FcγR and C1q binding while IgG1 and IgG3 correlated strongly (S3A Fig). This confirms that the IgG1 antibodies are likely responsible for the strong FcγR and C1q interaction which was associated with delayed disease progression. Several genetic signatures were previously identified to grant a degree of protection from HIV-1 disease progression [50]. Correlations between functional Fc features and time between SC and AIDS were significantly weaker when only individuals with no deletion in the CCR5 gene were included, but these correlations were minimally affected in individuals confirmed to not have the HLA-B57 or HLA-B27 genes (S3B Fig). Moreover, since there is a sizable global population that do not express the high affinity allelic variants of FcγRIIa and FcγRIIIa, we also included low affinity allelic variants of these receptors in our analyses. This did not affect the correlations between FcγRIIa and time between SC and AIDS, but for FcγRIIIa, there were significantly weaker or absent correlations with disease progression when we used the low affinity allelic variant (S3C Fig).

## Antibody Fc polyfunctionality and breadth of antibodies with functional Fc features are associated with delayed disease progression

Antibody Fc polyfunctionality has been associated with HIV-1 control [48]. Therefore, we aimed to predict Fc polyfunctionality for each individual in our cohort by calculating a Fc polyfunctionality score. This score was based on the capacity of their antibodies to interact with multiple activating FcγRs (FcγRI, FcγRIIa, FcγRIIIa, FcγRIIIb) and C1q, given that these

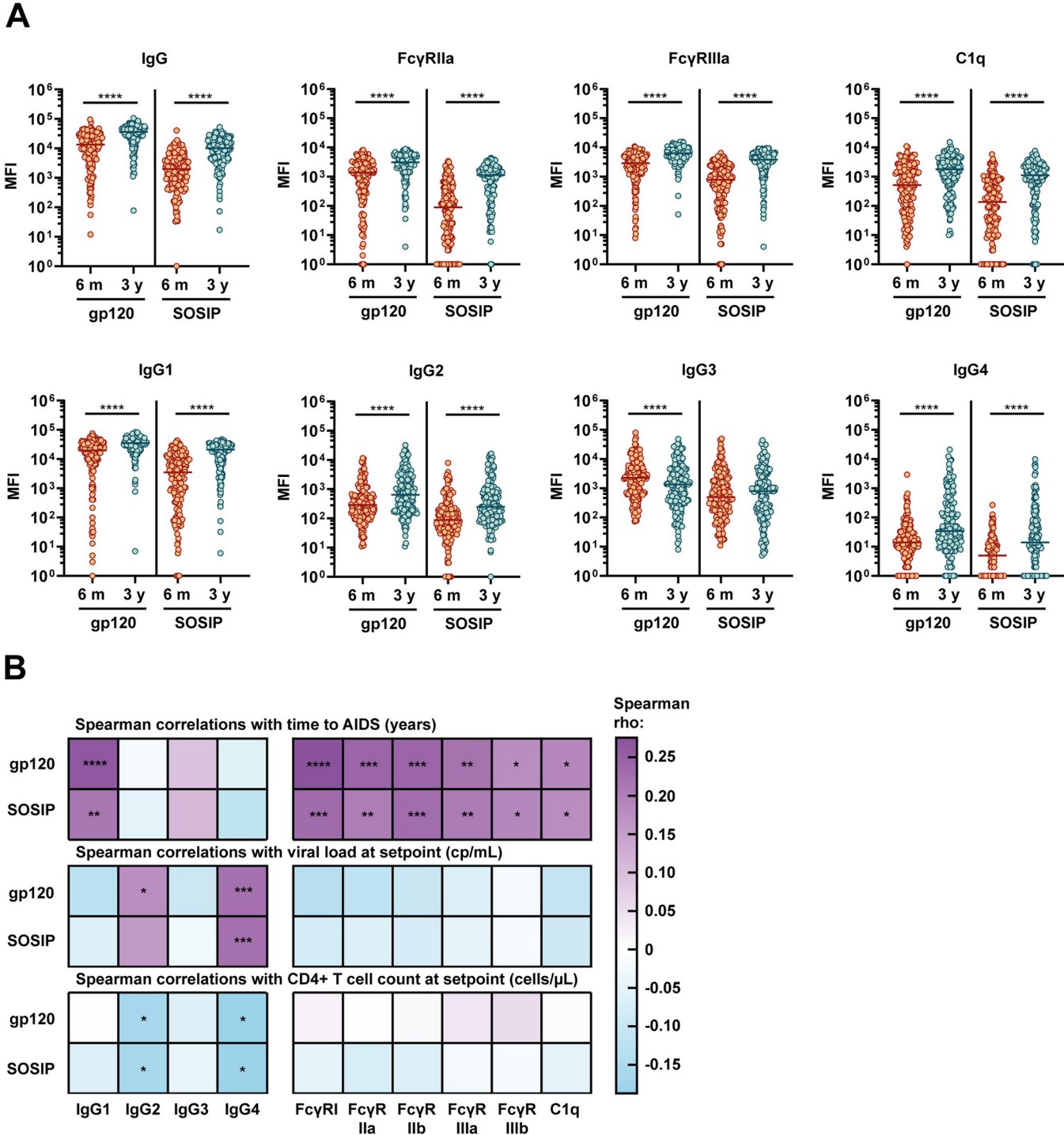

**Fig 1. Antibody features over time and associations with HIV-1 disease progression.** (A) 92BR020-specific antibodies targeting SOSIP and gp120 antigens in serum of IgG type, IgG1-4 subtype and interacting with FcγRIIa (high affinity allelic variant H131) (associated with antibody-dependent cellular phagocytosis), FcγRIIIa (high affinity allelic variant V158) (associated with antibody-dependent cellular cytotoxicity) and C1q protein (associated with complement activation and complement-dependent cytotoxicity). Antibody responses are compared between the time point 6 months after seroconversion (SC) and 3 years after SC using a Wilcoxon matched-pairs signed rank test. Only individuals with samples at both time points (n = 166) are included in this subfigure. The data is expressed as median fluorescence intensity (MFI) measured by Luminex assay. The eight features shown here are those we were most interested in based on literature, seven additional features are shown in S2A Fig. (B) Spearman correlation analysis between the antibody features analyzed in this study at 3 years post-SC and three clinical variables related to disease progression. Positive correlations between antibody features and time to AIDS or CD4+ T cell count at setpoint are related to delayed disease progression whereas positive correlations with viral load at setpoint are related to faster disease

progression. Color indicates the rho Spearman correlation coefficient and significant results are indicated by asterisks. The results were corrected for multiple comparisons using the Bonferroni-Holm method. Antibody features shown in this figure are for responses specific for gp120 and SOSIP proteins of subtype B strain 92BR020. Abbreviations: AIDS: acquired immunodeficiency syndrome; cp/mL: viral copies per milliliter. * = P<0.05, ** = P<0.01, *** = P<0.001, **** = P<0.0001.

interaction levels were previously demonstrated to highly correlate with a range of different Fc functions in cell-based assays [51–53]. There was a clear increase in Fc polyfunctionality in the groups with delayed disease progression at 3 years post-SC (Fig 2). In contrast, at 6 months post-SC, there was no association between Fc polyfunctionality and disease progression (S4 Fig).

Due to the immense variety among HIV-1 strains and frequent immune escape, broad immune responses are vital for protection from infection and disease progression [34]. To assess breadth, we generated six additional gp120 monomers and SOSIP trimers, amounting to a total of seven diverse strains of HIV-1: 94UG103 and BG505 (clade A), 92BR020 and JRCSF (clade B), 93IN905 and IAVIC22 (clade C) and 92TH021 (clade AE). Neutralization data was previously obtained for this cohort against six of the strains assessed in the current study (all except BG505) [55]. NS-EM analyses indicated that the SOSIP proteins form trimers, but some of the trimers formed aggregates or contained subpopulations of monomers and dimers (S1B Fig). However, this did not appear to influence the patterns of recognition by a panel of monoclonal antibodies specific for different HIV-1 epitopes (S1C Fig). At 6 months post-SC, 80–99% of participants in our cohort had IgG binding to the diverse gp120 antigens and 90–99% of participants had IgG binding to the different SOSIP Env proteins. At 3 years post-SC, 98–100% of participants had IgG specific for gp120 antigens and 99–100% had IgG against the SOSIP antigens (S5A Fig). In this cohort, 25% of participants were previously determined to have a bNAb response (defined as neutralizing >4 out of 6 strains with an $ID_{50}$>100 [55,56]). In contrast, virtually all participants had detectable IgG against gp120 as well as SOSIP antigens of all 7 strains used (S5B Fig). Gp120 and SOSIP-specific responses strongly correlated (S5C Fig). Moreover, IgG binding results at 3 years post-SC correlated significantly with the previously obtained neutralization potency results against the matched virus at the same time point (S5D Fig). Correlations were stronger for the SOSIP antigens compared to the gp120 antigens, which is consistent with the expectation that SOSIP antigens display more bNAb epitopes than gp120.

We calculated a breadth score across the seven gp120 antigens to assess the breadth of Fc features. For all features a trend was visible for increased breadth in individuals with slower disease progression at 3 years post-SC, congruent with higher breadth indexes (Fig 3). The effect was most pronounced for antibodies interacting with FcγRI and FcγRIIa. Effects were similar for SOSIP-specific responses (S6A Fig). In contrast, at 6 months post-SC, there was a trend for most participants with slower disease progression to induce FcγR binding antibodies that recognize a more narrow range of strains (S6B and S6C Fig). This was not observed for antibodies that interact with C1q protein.

## V3, gp41 and gp120-specific antibodies with functional Fc features correlate with delayed disease progression

Broad and potently neutralizing antibodies can target different epitope clusters of HIV-1 Env [57]. Therefore, we also analyzed whether antibody responses could recognize constructs presenting the following epitopes: the V1/V2 region (strain JRCSF), the V3 region (strain JRCSF), the CD4 binding site (derived from strain HXB2), gp41 (strain JRCSF) and the membrane-proximal external region (MPER, based on the 2E5 antibody epitope). In addition, we included

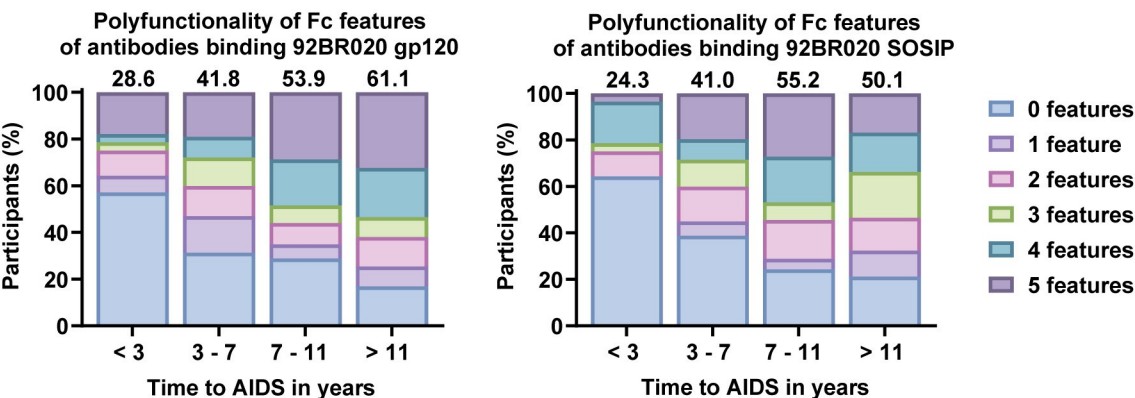

**Fig 2. Polyfunctionality of antibody Fc features.** Polyfunctionality was assessed by calculation of a polyfunctionality score at 3 years post-seroconversion (SC). We only assessed activating FcγR (FcγRI, FcγRIIa, FcγRIIIa, FcγRIIIb) and C1q interaction, and for each Fc feature we determined for each participant if there was a response that was higher than the median response of all responders in the cohort. The resulting scores (amount of features above the responder median, between 0 and 5, per person) were plotted for the four groups based on time between SC and acquired immunodeficiency syndrome (AIDS) (less than 3 years (n = 28), between 3 and 7 years (n = 147), between 7 and 11 years (n = 66) and more than 11 years (n = 71)) with one plot for 92BR020 gp120-specific responses and one plot for 92BR020 SOSIP-specific responses. In addition, a polyfunctionality index was calculated for each group as a quantitative measure of polyfunctionality. This is a value between 0 and 100 calculated by a weighted addition of the percentage of individuals in each category based on the number of features, derived from Larsen *et al.* [54], see methods). This index is shown above each bar. Polyfunctionality of antibodies against these two strains at 6 months post-SC are shown in S4 Fig.

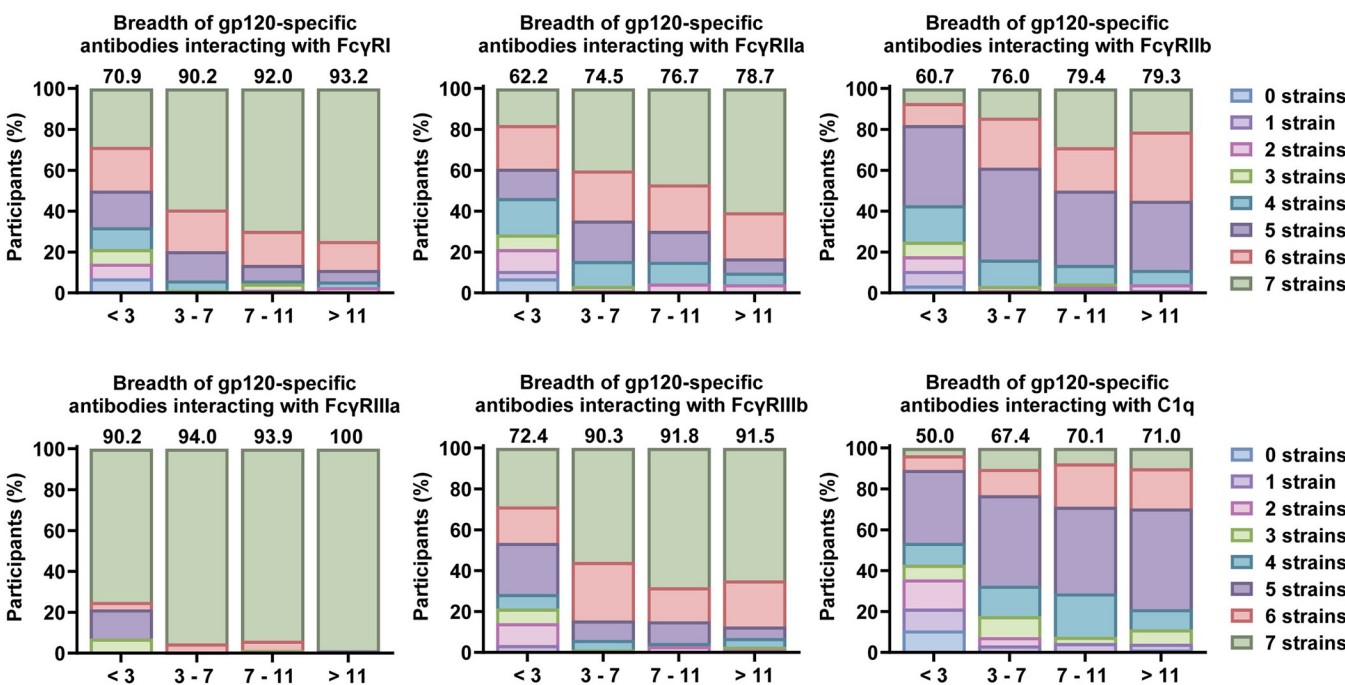

**Fig 3. Breadth of gp120-specific antibody interaction with FcγRs and C1q.** Breadth is compared between four categories based on time between seroconversion and acquired immunodeficiency syndrome (AIDS) diagnosis: less than 3 years (n = 28), between 3 and 7 years (n = 147), between 7 and 11 years (n = 66) and more than 11 years (n = 71). Bar charts show the percentage of individuals within each group with antibody interaction with the plotted FcγR or C1q above the responder cut-off for the color-coded amount of strains. Above each bar the breadth index for that group is shown as a quantitative measure of breadth. This is a value between 0 and 100 calculated by weighted addition of the percentage of individuals in each category based on the number of features, derived from Larsen *et al.* [54] (see methods). Breadth of gp120-specific antibody interaction at 6 months post-SC and breadth of SOSIP-specific antibody interaction at 3 years and 6 months post-SC are shown in S6 Fig.

a gp120 monomer fused to soluble CD4 (gp120-CD4; strain BaL) to assess antibodies that recognize CD4-induced (CD4-i) epitopes, which were previously associated with antibody-mediated effector functions [58]. These measurements were included only for IgG, IgA, IgM, IgG1-4, FcγRIIa, FcγRIIIa and C1q. Response rates for V1/V2, MPER and CD4bs constructs were low, while most participants had V3, gp41 and gp120-CD4 specific antibodies with many different features (S7 Fig). Spearman correlations showed that mainly IgG1 antibodies and antibodies interacting with FcγRs and C1q targeting the V3 region at 3 years post-SC were associated with delayed disease progression (Fig 4A). We found no evidence for specific associations with CD4-i antibodies, since the associations between gp120-CD4 construct-specific antibody features and time to AIDS were not stronger than between gp120-specific antibody features and time to AIDS. We also observed epitope-specific associations with setpoint viral load.

To further analyze the epitope specificity of the Fc functional response, we normalized the levels of antibodies interacting with FcγRIIa, FcγRIIIa and C1q across the six epitope scaffolds and one gp120 and SOSIP protein (belonging to clade B strain JRCSF) (Z-score). Antibody binding to these MPER and V1/V2 constructs was relatively low for antibodies interacting with FcγRIIa, FcγRIIIa and C1q, while JRCSF gp41-specific antibodies frequently interacted with FcγRIIa, FcγRIIIa and C1q (data included in the online repository). SOSIP- and JRCSF V3-specific antibodies interacting with C1q were slightly increased in individuals with slower disease progression (Fig 4B). At 3 years post-SC, there was also an increase in gp120-, gp120-CD4-, V3- and gp41 construct-specific antibodies interacting with FcγRIIa, while the largest differences were observed for FcγRIIIa, with increased levels of gp120-, gp120-CD4-, SOSIP- and V3 construct-specific antibodies in individuals with slower disease progression.

## A profile of antibody features for delayed disease progression

To obtain a comprehensive profile of antibody features associated with disease progression, we compiled all data at 3 years post-SC and applied feature selection (elastic net) followed by a partial least squares regression (PLSR) analysis with time to AIDS as an independent variable (Figs 5 and S8). The results showed that antibodies which target V3, gp41 and gp120 antigens and interact with a range of FcγRs as well as C1q are drivers for differences in the rate of disease progression in individuals with a longer time between SC and AIDS in this model, and that the main contributing antibody types are IgG and IgG1. Of the FcγRs, FcγRIIa had the largest role. Breadth was not selected, whereas polyfunctionality did assume a clear role in the model. We also included neutralization titers and neutralization breadth into the model, however, none of these were selected by elastic net. Thus, gp120-, V3- and gp41-specific polyfunctional IgG, strongly reacting with FcγRs and especially FcγRIIa, provides a profile associated with delayed disease progression. To further confirm the association with disease progression, we also performed an antibody-dependent cellular phagocytosis assay on a subset of participants (n = 140). ADCP positively correlated with time between SC and AIDS (r = 0.192, P<0.05). The ADCP score also significantly correlated with two features that were prominent in the PLSR analysis: gp120-specific antibody interaction with FcγRIIa and FcγRI (S8 Fig).

## Coordination between Fc features and antibody neutralization

Antibody-mediated effector functions were previously associated with the development of bNAbs [59]. Therefore, we also looked into the profile of features responsible for this association. Spearman correlation between geometric mean (geomean) neutralization titers (of all 6 strains tested) and 92BR020-specific Fc features identified that nearly all measured features (except IgM) at 3 years-post-SC were strongly correlated with neutralization breadth (S9A

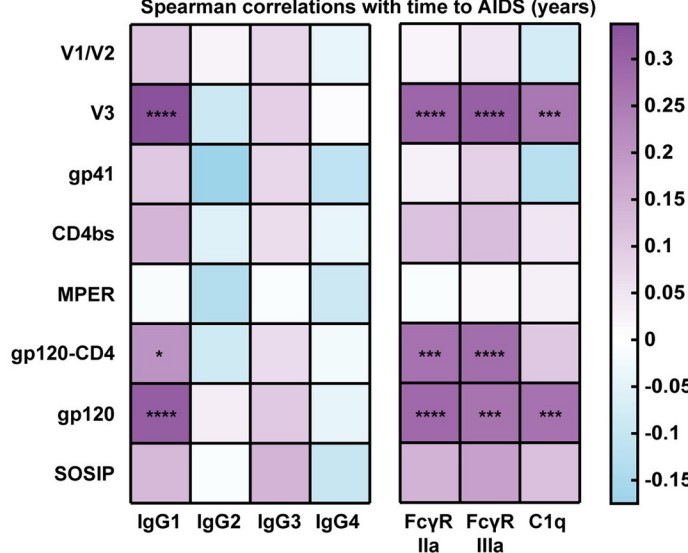

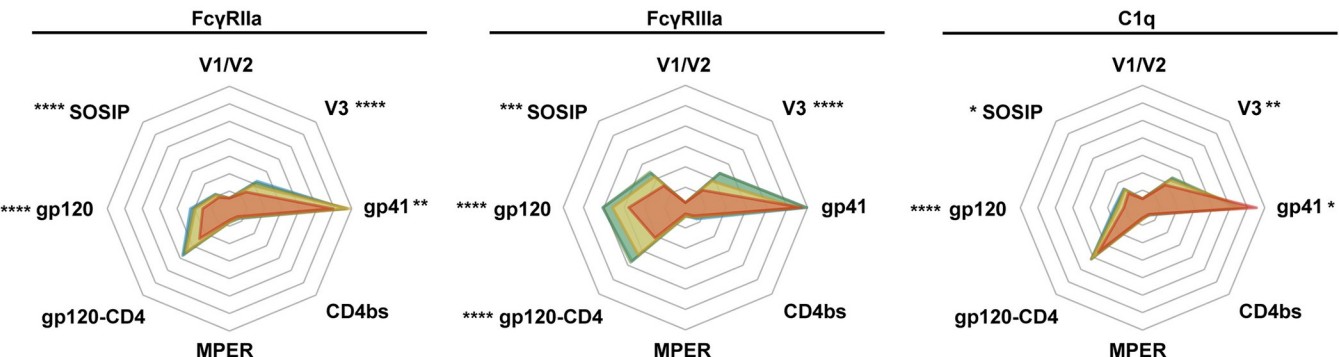

**Fig 4. Epitope specificity of antibodies interacting with FcγRs and C1q in different groups based on disease progression.** (A) Spearman correlation analysis between the antibody features analyzed in this study and time between seroconversion (SC) and AIDS diagnosis. The color indicates the rho Spearman correlation coefficient and significant results are indicated by asterisks. The results were corrected for multiple comparisons using the Bonferroni-Holm method. (B) Levels of antibodies were normalized by Z-score across the eight protein antigens to allow comparison of the specificity of the response of different types of antibodies and antibody interactions. This allows comparison of the different graphs even though the original magnitude of the different antibody types and functions was quite different. Therefore, the scale of the graph is not relevant, but the pattern and the ratio between different specificities can be compared between plots. In addition, Kruskall-Wallis tests were used to compare the responses to each protein antigen separately and the results are shown on the figure as stars. The plots are color-coded for four categories based on time between SC and AIDS diagnosis: less than 3 years (n = 28), between 3 and 7 years (n = 147), between 7 and 11 years (n = 66) and more than 11 years (n = 71). Protein antigens shown in this figure are gp120 and SOSIP proteins of subtype B strain JRCSF, V1/V2, V3 and gp41 epitope constructs were also based on the JRCSF sequence. Constructs comprising the CD4 binding site (CD4bs), the MPER and a covalently linked fusion of gp120 and CD4 (gp120-CD4) were previously described (see methods). * = P<0.05, ** = P<0.01, *** = P<0.001, **** = P<0.0001.

Fig). Additionally, most epitopes were part of this association as indicated by the many significant results in the Spearman correlations with the epitope constructs (S9B Fig). PLSR analysis after elastic net feature selection implicated a dominance of heterologous SOSIP-specific

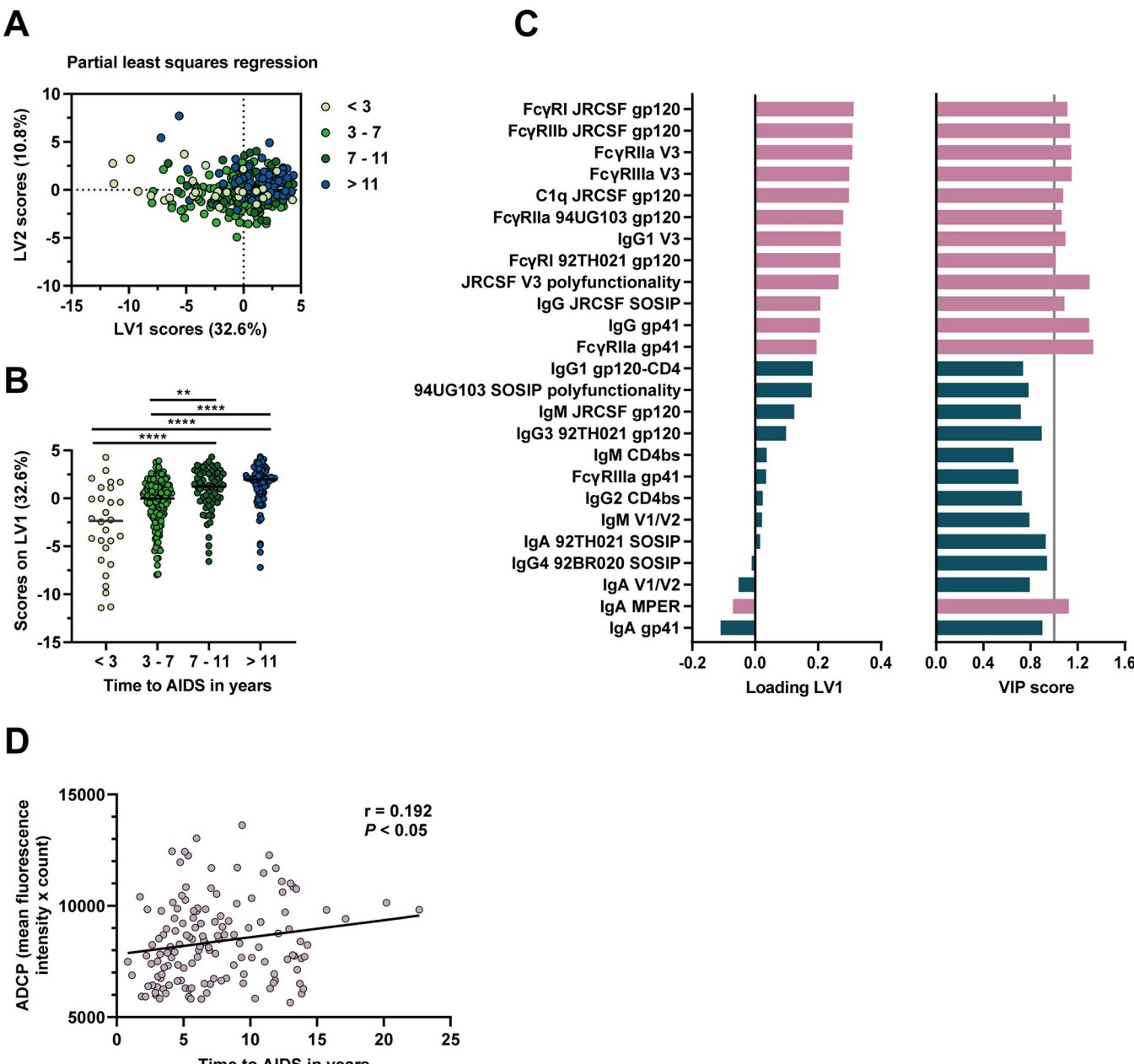

**Fig 5. Further identification and confirmation of associations with HIV-1 disease progression.** A partial least squares regression analysis was performed with time between seroconversion (SC) and acquired immunodeficiency syndrome (AIDS) diagnosis in years as the independent variable. Twenty-five Fc features were included in the model as selected by elastic net feature selection. Data for the feature selection included antibody responses for all gp120, SOSIP and epitope antigens including antibody types, subtypes, interaction with all FcγRs and C1q, breadth scores, polyfunctionality scores, neutralization titers and neutralization breadth. Further properties of the model are shown in S8 Fig. (A) The model comprised two latent variables (LVs) and they are plotted against each other. (B) LV1 is displayed in a column plot with four groups based on time to AIDS (less than 3 years (n = 28), between 3 and 7 years (n = 147), between 7 and 11 years (n = 66) and more than 11 years (n = 71)) compared using a Kruskal-Wallis test followed by a Dunn's multiple comparison test. (C) Loading and variable importance in projection (VIP) scores of all 25 included features are shown, with variables with a VIP score higher than 1 shown in pink (also indicated by the grey vertical line) and the remaining variables shown in dark green. (D) Spearman correlation analysis between antibody-dependent cellular phagocytosis (ADCP) of 92BR020 gp120-conjugated beads by THP-1 cells at 3 years post-SC and time between SC and AIDS. This analysis was done on a subset of 140 participants with a similar distribution of time between SC and AIDS as the full cohort. r = rho Spearman correlation coefficient, ** = P <0.01, *** = P<0.001, **** = P<0.0001.

responses, with both IgG levels and interaction with Fc γ receptors, especially FcγRIIa, positively associated with geomean neutralization titers (S9C–S9E Fig). Interestingly, when we performed associations with the features measured early in infection at 6 months post-SC, we also observed many associations with broad neutralization activity at 3 years post-SC, indicating a possible predictive effect of early Fc features for development of broad neutralizing activity (Fig 6A). Targeting of gp120 and SOSIP by antibodies interacting with all FcγRs and C1q strongly correlated with geomean neutralization titers. Moreover, targeting of V3, gp41 and the CD4 binding site by IgG1 and antibodies interacting with FcγRIIa and FcγRIIIa were also associated with the development of broad neutralizing activity (Fig 6B). Individuals with high neutralization titers and larger neutralization breadth had more Fc polyfunctional antibody responses at both time points (S10 Fig). PLSR analysis again implicated heterologous SOSIP-specific antibodies of the IgG type and interacting mainly with FcγRIIa, but also C1q, to be important for this association (Fig 6C and 6D).

## Discussion

Evidence is accumulating that antibody-mediated effector functions can play a considerable role in the disease course of HIV-1 infection, as well as in other viral diseases [60, 61]. Polyfunctional antibody responses, driven by IgG1 and IgG3 subclasses and with low levels of the functionally inferior IgG2 and IgG4 subclasses, were previously associated with protection from disease progression [8,9,47,48]. In this study, we assessed an array of antibody Fc features to pinpoint specific biophysical antibody features that are associated with delayed disease progression at two time points in a large, well-defined cohort. This data can inform antibody-based strategies for HIV-1 prevention, therapy and cure. Significant univariate correlations were observed for IgG1 antibodies interacting with many FcγRs and C1q. Multivariable regression analyses confirmed these associations and marked an important role for the polyfunctionality of the antibody response.

An association between Fc polyfunctionality and delayed disease progression has been described in earlier studies, even in absence of univariate associations of individual functions[48]. Considering the many individual associations in our cohort, it is not surprising that polyfunctionality takes a strong position in our final profile. In our study, antibodies with functional Fc features were also broader in participants with delayed disease progression. This indicates that these antibodies may recognize more conserved epitopes. This may be beneficial because this may allow protection against evolving viral strains within chronically infected individuals. These conserved epitopes may be too important to allow for major antigenic change, which often occurs on strain-specific epitopes. However, our multivariate regression analysis identified that the role of this antibody breadth is likely smaller in comparison to other factors such as specific antibody features as well as Fc polyfunctionality. The final profile of specific biophysical antibody features includes C1q and various FcγRs, with the largest emphasis on FcγRIIa. This suggests that phagocytosis may be important in control of HIV-1 and thus next generation antibody-based (curative) therapeutics should aim to include this antibody function. This finding was further supported by confirmation of the association between ADCP and disease progression in our cohort. The dominance of IgG1 antibodies, the association of IgG1 with delayed disease progression and the observed associations between IgG1 and antibody features such as FcγRIIa interaction, indicate that IgG1 is likely most responsible for this effect in our cohort. Interestingly, the inhibitory FcγRIIb is also part of the profile of antibody features associated with reduced disease progression, indicating that a balanced response may also be important. Moreover, this is also interesting since it was shown that FcγRIIb responses may be associated with improved affinity maturation in the context of Influenza virus [62,63].

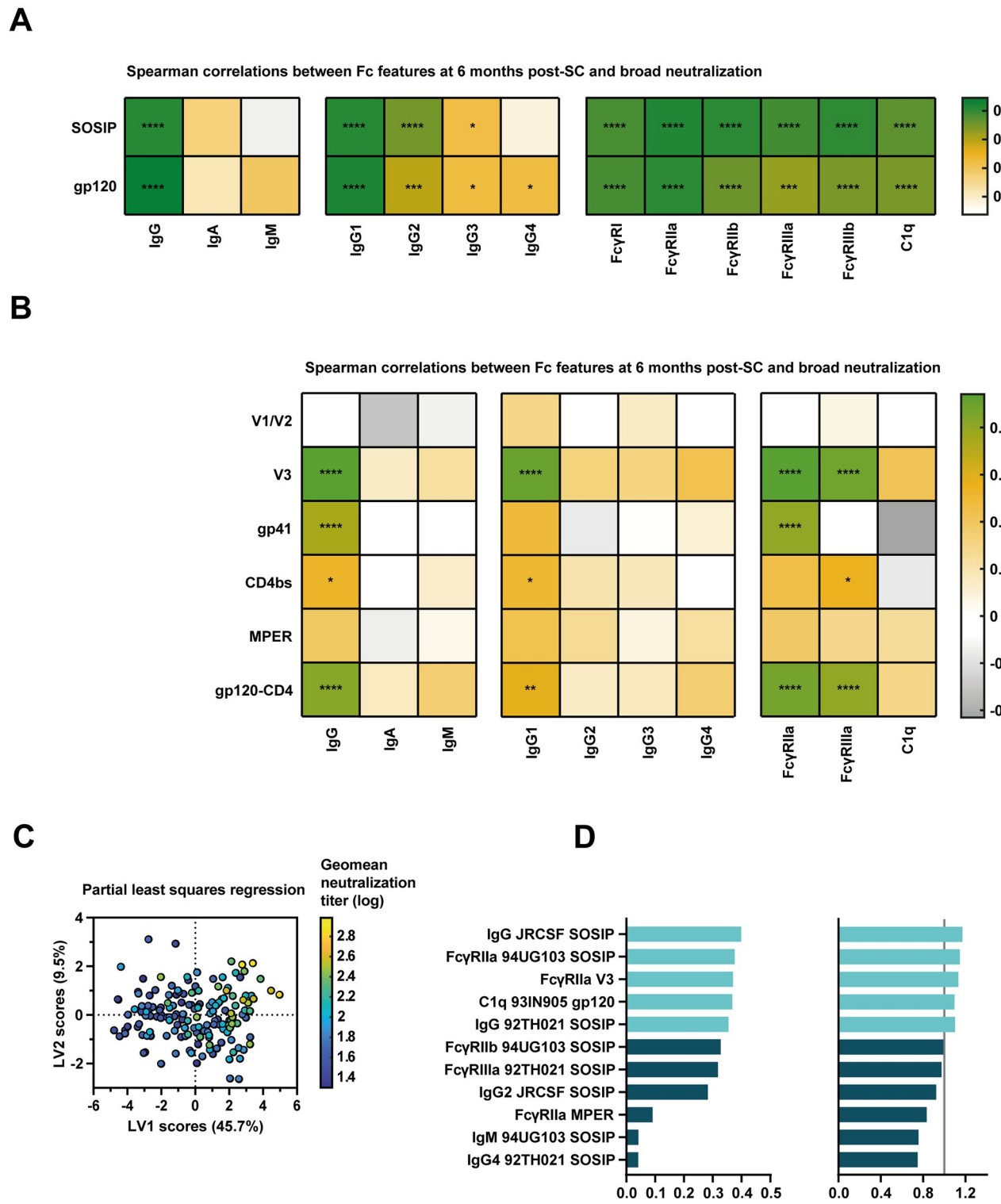

**Fig 6. Associations between antibody Fc features and broad HIV-1 neutralization activity.** (A) Spearman correlation analysis between the antibody features analyzed in this study at 6 months post-seroconversion (SC) and geometric mean (geomean) half-maximal infective dilution (ID$_{50}$) pseudovirus neutralization titers at 3 years post-SC (n = 166). Antibody features shown in this panel are for responses specific for gp120 and SOSIP proteins of subtype B strain 92BR020. Correlations with antibody Fc features at 3 years post-SC are shown in S9 Fig. The color indicates the rho Spearman correlation coefficient and significant results are indicated by asterisks. The results were corrected for multiple comparisons using the

Bonferroni-Holm method. (B) Spearman correlation analysis between the antibody features analyzed in this study at 6 months post-SC and geomean $ID_{50}$ pseudovirus neutralization titers at 3 years post-SC (n = 166). Antibody features shown in this figure are for responses specific for gp120 and SOSIP proteins of subtype B strain JRCSF, V1/V2, V3 and gp41 epitope constructs were also based on the JRCSF sequence. Constructs comprising the CD4 binding site (CD4bs), the membrane-proximal external region (MPER) and a covalently linked fusion of gp120 and CD4 (gp120-CD4) were previously described (see methods). The color indicates the rho Spearman correlation coefficient and significant results are indicated by asterisks. The results were corrected for multiple comparisons using the Bonferroni-Holm method. (C) Partial least squares regression analysis geomean $ID_{50}$ pseudovirus neutralization titer as independent variable. Eleven Fc features were included in the model as selected by elastic net feature selection with alpha 0.5 and 16-fold cross-validation. The model comprised two latent variables (LVs), had a $R^2$ of calibration of 0.389, root mean square error of calibration of 0.267 and root mean square error of cross-validation of 0.336. (D) Loading and variable importance in projection (VIP) scores of all 11 included features, with variables with a VIP score higher than 1 shown in light blue (also indicated by the grey vertical line) and the remaining variables shown in dark blue. * = P<0.05, ** = P<0.01, *** = P<0.001, **** = P<0.0001.

The correlation coefficients that we found between antibody Fc properties and disease progression were encouraging (highest r = 0.34), considering that several previous studies found either weak associations or did not find any correlation with envelope-specific effector functions at all [8,48,64]. While some studies did find higher correlation coefficients compared to our study, those results were based on current viral loads measurements at unknown time points and may be substantially influenced by the time of sampling [45,46]. In comparison, the current study presents a uniquely detailed profile of aspects of the antibody response that are associated with disease progression. Interestingly, similar correlation coefficients were found in the setting of subtype C infection with a comparable sample size, suggesting that our findings may not be limited to subtype B infections [8]. Collectively, these studies contribute to the growing evidence for the link between antibody-mediated effector functions and protection against disease progression. However, these studies are unable to prove causality and therefore it remains possible that these are indirect correlations, linked with other factors that do play a mechanistic role in the observed associations. Moreover, many additional factors have been associated with delayed disease progression, including immune activation, T cell immunity, properties of the infecting virus strain as well host factors such as gene polymorphisms and psychosocial factors [65]. Future studies are needed to further unravel this complex network of associations and to clarify if antibody-mediated effector functions play an active role in protection from disease progression.

When investigating associations of antigen-specific antibody subclasses and disease progression, we observed strong correlations between IgG1 and FcγR interaction while this was not seen for IgG3. This suggests that the majority of FcγR interactions we observed are mediated by IgG1, despite IgG3 being considered the most functional subclass [66]. This is likely due to the loss of IgG3 during the 3 years after infection. Indeed, 2.5 years following the 6 month time point, we observed that all types of antibodies increased in frequency, while gp120-specific IgG3 decreased. IgG3 is the first subclass on the immunoglobulin locus and therefore IgG3 is the subclass most likely to wane first [29]. Since Fc functionality in absence of IgA was associated with the limited protection in the RV144 HIV-1 vaccine trial [67] and IgG2 and IgG4 levels were previously associated with reduced Fc functionality in the VAX0003 HIV-1 vaccine trial [68, 69], we were interested to see if we could also observe differences related to disease progression. We did observe negative effects related to disease progression for IgG2 and IgG4 antibodies, but associations for IgA were not frequent in our analysis.

The SOSIP trimers included in this study were designed to maximize binding of neutralizing antibodies and limit binding by non-neutralizing antibodies [70]. Therefore, the stronger associations found for gp120 monomers compared to SOSIP trimers are likely caused by both non-neutralizing and neutralizing antibodies. We found that antibodies with functional Fc features, especially when interacting with C1q, more often recognize CD4-i epitopes and the V3 region. The V3 region is a well-known target for both neutralizing and non-neutralizing

antibodies with effector functions [71]. CD4-i antibodies can also mediate broad and potent effector functions [58,72]. However, CD4-i antibodies were rarely more associated than gp120-specific antibodies in our univariate correlation analyses, indicating the lack of a specific relationship with disease progression in our cohort. V3 and gp41 were also frequent targets of antibodies that associate with delayed disease progression. When HIV-1 buds from infected cells, gp120 can be shed from the cell surface, leaving gp41 stumps. Gp41-specific antibodies mediating effector functions can clear these infected cells, which may underlie their possible role in the control of disease progression [73]. Importantly, there were large differences between the quantity of antibodies recognizing the different epitope constructs. Therefore, the strength and statistical significance of the associations for the different epitopes are likely influenced by the relative abundance of antibodies with certain specificities. Associations for highly abundant antibody specificities may be overemphasized while it is also plausible that we missed potential associations because of the lower titers for V1/V2, the CD4bs and the MPER. We also observed a difference in antibody binding between strains, therefore the choice of strain for the epitope constructs also likely affects the results. Consequentially, this data does not allow definite conclusions on the specificity of the antibodies that are most important for the association with delayed disease progression.

That antibodies can both mediate effector functions and have neutralizing activity is now established, yet there is currently not much emphasis on measuring the induction of antibodies with both functionalities. In our analysis, we identified that early antibody responses including IgG1, antibodies with the capacity to engage FcγRIIa, FcγRIIIa and C1q are predictive for the development of antibodies with broad neutralization at 3 years post-SC. This complements a previous study which found that polyfunctional antibodies, FcγR binding antibodies, antibody-dependent cellular trogocytosis and antibody dependent complement deposition early in HIV-1 infection were associated with the development of neutralization breadth [59]. It was suggested that this indicates possible shared underlying factors for both Fc functionality and neutralization breadth. We also observed an unexpected association between neutralization breadth and antibodies targeting V3 and gp41. In our study, early heterologous SOSIP-specific responses were also predictive for broad and potent neutralization activity, but these responses did not associate with disease progression. It is plausible that the strong correlations we observed between antibody Fc features and neutralizing antibodies at 3 years post-SC can be explained by the IgG1 titers in our cohort. However, the previously established lack of an association between (broad) neutralization and delayed disease progression, indicates that the Fc functional properties of this antibody response are likely crucial for the observed association with delayed disease progression.

An important cause of heterogeneity in population immunity to HIV-1 is that there are two allelic variants of both FcγRIIa and FcγRIIIa [74]. In some populations, the allelic variants of FcγRIIa and FcγRIIIa with lower affinity for IgG are as prevalent as the high affinity variants [21]. We performed most of our analyses with the high affinity allelic variants, as is common practice, but did include a sub-analysis with the low affinity allelic variants. This did not change the outcomes for FcγRIIa, which is fortunate since FcγRIIa did play a larger role in the outcome related to delayed disease progression. However, for FcγRIIIa there was a marked lowering of the correlation coefficients as well as the statistical significance of the associations. Antiviral strategies are more useful when they work also for the part of the population that have the allelic variants with low affinity for IgG. Therefore, this outcome should be confirmed in future studies. Other studies have reported higher correlation coefficients between antibody functions and disease progression when analyzing only individuals with protective alleles such as HLA-B57 [46]. We did not find substantial impact of HLA-B57 and HLA-B57 alleles on our results, but the associations in our cohort were less pronounced when participants with the

protective 32 base pair deletion in CCR5 were excluded. This demonstrates the importance of including analyses of these genetic signatures to allow correct interpretation of the results.

We did not identify many associations between Fc features and disease progression at 6 months post-SC. It is likely that the smaller sample size at the 6 month time point affected these results. However, some interesting trends were observed at 6 months post-SC: we found no link between antibody Fc polyfunctionality and disease progression at this time point and we found trends indicating that having broader and less functional antibodies could actually be predictive for faster disease progression. It is possible these trends are related to a higher viral load during early infection in participants with fast disease progression, as has been suggested before [4]. However, we did not find any significant correlations with setpoint viral load at 6 months post-SC. These trends suggest that further research on this early response in larger cohorts may be of interest. Unfortunately, inclusion of individuals in cohorts this early after seroconversion is uncommon.

The emphasis on FcγRIIa in the final profile of antibody features related to disease progression and the observed correlation between ADCP and delayed disease progression suggest that phagocytosis may be important for the control of HIV-1. However, FcγRIIa can also contribute to antigen presentation and other Fc functions including antibody mediated trogocytosis, cytotoxicity and cytokine/enzyme release [75–77]. Therefore, FcγRIIa interaction can be a useful template for the design of novel antibody-based (curative) therapeutics. For example, using antibody engineering strategies that have been reported to increase affinity for FcγRIIa [78]. In such an approach, the affinity for inhibitory FcγRIIb should likely also be considered to achieve a balanced response. Moreover, therapeutic antibodies can be targeted to specific epitopes, such as the gp41 and V3 epitope clusters that we identified to be associated with HIV-1 control. However, passive administration of antibodies have shown that neutralizing, but not non-neutralizing antibodies are associated with delayed viral rebound. Therefore, ideally these antibodies should additionally be broadly neutralizing. Moreover, methodology similar to this study could allow characterization of antibody responses induced by infection and/or vaccination, which can confirm whether these antibody responses have a desirable profile in terms of antibody-mediated effector functions.

In conclusion, our study provides a specific biophysical profile of antibody features attractive for preventative and therapeutic strategies against HIV-1. High levels of broad and polyfunctional IgG1, low levels of IgG2 and IgG4 as well as high levels of interaction with all human FcγRs and especially FcγRIIa were identified in this study as correlates of delayed disease protection. Moreover, antibodies with functional Fc features were also predictive of the development of potent and broad neutralizing antibodies. Thus, our results demonstrate the benefit of a broad and polyfunctional antibody response in HIV-1 infection. Strategies adopting the information presented here may provide interesting avenues for further development of alternative therapeutics and HIV-1 cure strategies.

## Methods

### Ethics statement

The study was approved by the institutional medical ethtics committee of the Amsterdam Medical Center and the ASC is conducted in accordance with the ethical principles in the declaration of Helsinki. Prior to data collection, informed consent was obtained in writing.

### Study population

The study population has been described previously [55,79] and consisted of 382 individuals enrolled between October 1984 and March 1986 to the Amsterdam Cohort Studies on HIV

infection and AIDS (ACS). All individuals were exclusively infected with HIV-1 subtype B. None of the participants received antiretroviral therapy during the study period (up until 3 years post-seroconversion). Seroconversion (SC) dates were documented or imputed as part of the cohort as described in previous studies [3,79,80]. AIDS diagnosis in absence of treatment was known for 268 individuals and 44 individuals were not diagnosed with AIDS during their participation in the cohort, but participated in absence of treatment for a least 11 years.

For these 382 individuals, serum samples were selected of approximately 3 years post (imputed) SC date (n = 382) and approximately 6 months post (imputed) SC date (n = 166). BNAb responses were previously determined in the same cohort, also at 3 years post (imputed) SC [55] against six tier 2 HIV-1 pseudoviruses from different clades by Monogram BioSciences (92BR020, 92TH021, 93IN905, 94UG103, IAVIC22, JRCSF) [81]. These viruses were previously shown to cover 93% of variation in a larger 15-virus panel [56]. Information on time between SC and AIDS or between SC and study drop-out in years, viral load at setpoint and CD4+ T-cell counts at setpoint, was obtained from the cohort database. These are described in Tables 1 and S1, no matching was performed. Some participants stopped participating in the cohort or started antiretroviral therapy before AIDS diagnosis. Individuals who participated 11 years or more without antiretroviral therapy and with no AIDS diagnosis were included in the analyses related to time to AIDS.

## Design of protein antigens

Monomeric gp120 subunits of HIV-1 envelope corresponding to strains 92BR020, 92TH021, 93IN905, 94UG103, IAVIC22, JRCSF and BG505 were designed with a L111A mutation to prevent dimer or trimer formation and a truncation at position 512 (HXB2 numbering) as described [82]. Native-like SOSIP stabilized pre-fusion HIV-1 envelope glycoprotein trimers corresponding to strains 92BR020, 92TH021, 93IN905, 94UG103, IAVIC22, JRCSF and BG505 were designed by incorporating a panel of amino acid changes previously described as SOSIP v.7 [83]. The gp41 construct was designed from a gene corresponding to amino acids 543–665 in HXB2 numbering of JRCSF Env as described earlier [84] followed by a hexahistidine (his) tag. The V1/V2 and V3 regions were incorporated on a gp70 scaffold sequence with his tag as previously described by others [85]. Incorporated sequences were amino acids 291–336 in (HXB2 numbering) of the V3 region from JRCSF Env and 120–204 in (HXB2 numbering) of the V1/V2 region from JRCSF Env. All construct sequences were ordered from Integrated DNA Technologies and cloned into a pPPI4 plasmid.

## Production and purification of protein antigens

All protein antigens were produced in HEK293F cells (Invitrogen) cultured in Freestyle medium (Life Technologies). Plasmids were transfected with polyethylenimine hydrochloride (PEI) MAX (Polysciences) at 1 mg/L and plasmids 312.5 μg/L in a 3:1 ratio in 50 mL Opti-MEM (Gibco) per liter. For SOSIP proteins, the envelope plasmid was co-transfected with furin (4:1 ratio). Supernatants were harvested 7 days post transfection by centrifugation at 4000 rpm for 30 min followed by filtration of the supernatant using 0.22 μM Steritop filter units (Merck Millipore). Gp120 protein was purified from the filtered supernatant using Sepharose affinity columns conjugated with 2G12. SOSIP protein was purified from the filtered supernatant using Sepharose affinity columns conjugated with PGT145, 2G12 or Gineaus Nivalis Lectin (GNL). His-tagged proteins were purified with affinity chromatography using NiNTA agarose beads (QIAGEN). Next, proteins were further purified using size exclusion chromatography on a Superdex200 10/330 G/L column (GE healthcare) to purify the monomer fraction only (for gp120 protein and epitope scaffolds) or the trimer fraction only

(for SOSIP proteins). The desired fractions were pooled and concentrated using Vivaspin centrifugal concentrators (Sartorius). The newly produced SOSIP proteins were characterized using blue native PAGE, negative-stain electron microscopy, and by assessing binding of a panel of well-characterized monoclonal antibodies. The CD4 binding site construct we used is called the resurfaced stabilized core 3, which is a stabilized gp120 core with all known bNAb epitope clusters removed or mutated, except for the CD4 binding site [86]. This construct was produced and purified the same way as the gp120 proteins.

## Production of Fc gamma receptor ectodomain dimers

Plasmids for the ectodomain dimers of human FcγRIIa-H131 and FcγRIIIa-V158 (high affinity allelic variants) were previously described [87]. The low affinity allelic variants FcγRIIa-R131 and FcγRIIIa-F158 were generated by mutagenesis using the QuikChange II Site-directed mutagenesis kit (Agilent Technologies) using the manufacturer's protocol. They were produced in HEK293F cells as described above. Human FcγRIIb was produced in a similar manner. Plasmids contained the genes followed by a his-tag and AviTag. The proteins were purified with affinity chromatography using NiNTA agarose beads. Protein was labeled with biotin on the AviTag with birA ligase (Genecopoeia) followed by size exclusion chromatography as described above.

## Biotinylation and Streptavidin-PE conjugation of Fc gamma receptor tetramers and human C1q

Human FcγRI and FcγRIIIb proteins were acquired from Invitrogen and human purified C1q protein was acquired from Complement Technologies. Proteins were biotinylated using the EZ-Link Sulfo-NHS-LC-Biotinylation Kit (Thermo Fisher Scientific) following the manufacturer's protocol. On the same day the assay was performed, the biotinylated proteins were conjugated to Streptavidin-PE (Thermo Fisher Scientific) based on molar amount and incubated 30 minutes at room temperature.

## Native PAGE analysis and Coomassie staining

SOSIP and gp120 envelope proteins were mixed with loading buffer ((1M MOPS + 1M Tris, pH 7.7)+ 1000μl 100% Ultrapure Glycerol (Invitrogen cat#15514–011) + 50μl 5% Coomassie Brilliant Blue G-250 + 600μl milli-Q water) and loaded in a 4–12% Bis-Tris NuPAGE gel (Invitrogen). The gel was run for 1,5 hour at 200 V at 4˚C using Anode-Buffer (20x NativePAGE Running Buffer (Invitrogen) in milli-Q water) and Cathode-Buffer (1% NativePAGE Cathode-Buffer Additive in Anode-Buffer, Invitrogen). The gel was stained using a QC colloidal Coomassie stain (Bio-Rad).

## Negative-stain electron microscopy

SOSIP proteins were diluted in Tris-buffered saline to ~0.02 mg/mL and adsorbed onto carbon-coated copper mesh EM grids. Excess solution was blotted using filter paper and the grids were subsequently stained for 45 s using a 2% (w/v) solution of uranyl formate, following by blotting. Approximately 100 micrographs per sample were collected using an FEI Tecnai Spirit T12 transmission electron microscope equipped with an FEI Eagle 4K CCD camera. Automated data collection was performed using Leginon [88], and subsequent data processing steps, including 2D classification, were done using Relion 3.0 [89]. Estimation of native-like trimers was done by comparing 2D classes to previously published images of well-folded SOSIP trimers [90,91]. Classes containing smaller (less massive) particles were labeled as

"dimer/monomer", while larger particles not clearly belonging to either "native-like" or "dimer/monomer" were labeled as "malformed."

## Coupling of HIV-1 proteins to Microspheres

HIV-1 and control antigens were covalently coupled to Magplex microspheres (Luminex) using a two-step carbodiimide reaction. In addition to the constructs for which production and purification was described above, we received two HIV-1 antigens through collaborators. The full-length single-chain, a chimeric construct composed of BaL gp120 fused with a flexible linker to the N terminus of the outer domains of CD4, was kindly provided by George K. Lewis from the University of Maryland School of Medicine [92]. A scaffold based on the epitope of MPER antibody 2E5 generated by rational computational engineering was kindly provided by William R. Schief from the Scripps Research Institute [93]. Microspheres were washed with 100 mM monobasic sodium phosphate pH 6.2, activated by addition of Sulfo-N-Hydroxysulfosuccinimide (Thermo Fisher Scientific) and 1-Ethyl-3-(3-dimethylaminopropyl) carbodiimide (Thermo Fisher Scientific) and incubated for 30 minutes at room temperature on a rotator. Microspheres were washed 2x with 50 mM MES pH 5.0 and the antigens were added in a ratio of 75 μg SOSIP trimer to 12,5 million beads, at equimolar concentration for other trimer antigens and at 3x equimolar concentration for monomeric antigens. Microspheres and antigen were incubated for 3 hours on a rotator at room temperature. Microspheres were blocked for 30 minutes with PBS containing 2% bovine serum albumin (BSA), 3% fetal calf serum (FCS) and 0.02% Tween-20 at pH 7.0. Finally, the microspheres were washed and stored at 4°C in PBS containing 0.05% sodium azide. Tetanus toxoid (Sigma) and RSV-F protein (design and production described in [94]) were included as positive controls and BSA-blocked microspheres with no protein as a negative control.

## Luminex array for 15 antibody features

In black 384-well plates (Greiner Bio-One), 25 μl of microsphere mixture containing 1000 of each bead and 25 μl of diluted plasma were combined and incubated overnight on a shaker at 4°C. Plates were washed with PBS + 0.05% Tween-20 and then detector was added. For the detectors mouse anti-human IgG-PE (Southern Biotech), mouse anti-human IgG1-PE, mouse anti-human IgG2-PE, mouse anti-human IgG3-PE, mouse anti-human IgG4-PE, this was added at 1 μg/mL and incubated for 2 hours. For biotinylated mouse anti-human IgM (mAb MT22; MabTech) and biotinylated mouse anti-human IgA (MabTech) also 1 μg/mL was added, plates were incubated for 2 hours and afterward washed with PBS + 0.05% Tween-20 and incubated for 1 hour with streptavidin, R-Phycoerythrin conjugate (SAPE, Invitrogen). For FcγRIIa-H, FcγRIIa-R, FcγRIIIa-V, FcγRIIIa-F and FcγRIIb, 1.3 μg/mL was added and incubated for 2 hours followed by washing with PBS + 0.05% Tween-20 and incubation for 1 hour with SAPE. For biotinylated and SAPE conjugated C1q, FcγRI and FcγRIIIb, 1.3 μg/mL was added and incubated for 2 hours. Finally, plates were washed with PBS + 0.05% Tween-20, microspheres were resuspended in sheath fluid (Luminex) and measured on the Flexmap 3D instrument (Luminex).

In prior optimization experiments, a panel of sera from people with HIV-1 were titrated for all beads to determine the optimal dilution factor to be used for all detectors. The dilution was set at a workable dilution close to dilution estimated to give 50% of the maximum signal in the assay, or at minimum 1:100. This resulted in a dilution factor of 1:100 for IgG2, IgG3, IgG4, C1q, IgA, FcγRIIb and FcγRIIIa-F; 1:200 for IgM, FcγRIIIa-V, FcγRIIIb, FcγRIIa-R; 1:500 for IgG1, FcγRI, FcγRIIa-H; and 1:5000 for IgG.

From the resulting median fluorescence intensity (MFI) values, the MFI values from microspheres and buffer only wells were subtracted. All plates included titrations of sera from two persons with HIV-1, HIVIG (obtained through the NIH HIV Reagent Program) and a cocktail of bNAbs to allow normalization between plates. All plates contained 10 negative control sera from healthy donors from the same geographical area and time period. All assays were performed in replicate on different days. For characterization of the proteins on the beads, HIV-specific monoclonal antibodies were titrated for all beads and the area under the curve was calculated from the resulting MFI values.

### Antibody-Dependent Cellular Phagocytosis (ADCP)

Fluorescent Neutravidin beads (Invitrogen) were incubated with biotinylated 92BR020 gp120 (5μg per 10 μl beads) overnight at 4°C. Beads were subsequently spun down and washed twice in PBS containing 2% BSA to remove unbound antigen and block the remaining hydrophobic sites on the microspheres. The coated beads were diluted 1:500 in PBS 2% BSA and 50 μl of the bead suspension was added to each well of V-bottom 96-well plates. Serially diluted sera were added and the plates were incubated for 2 hours at 37°C. After incubation, plates were washed and $5 \times 10^4$ THP-1 effector cells (ATCC) were added to each well in a volume of 100 μl. Subsequently, plates were shortly spun down to promote contact between beads and cells and then incubated for 5 hours at 37°C. After incubation, the cells were washed, resuspended in PBS 2% FCS and analyzed by flow cytometry. Phagocytic activity was defined as the Fluorescein isothiocyanate (FITC) mean fluorescence intensity multiplied by the beads positive cells.

### Data analysis

We determined a cut-off for a detectable antigen-specific antibody response for each antigen individually using sera of 10 HIV-1-negative individuals and the no protein control bead. This was as follows: a response above 100 MFI, more than 3x the MFI of the no protein control bead in the same well and above the 95th percentile of the measurements of the 10 HIV-1-negative individuals for the same antigen. One major outlier HIV-1-negative individual had to be excluded from these calculations for IgG3. The sera of the HIV-1-negative individuals were always included on the same plate as the cohort samples. Responder cut-offs were the same for the samples at 6 months and 3 years post-SC.

Polyfunctionality scores were calculated as follows. Only functional Fc features were included: interaction with activating FcγR (FcγRI, FcγRIIa, FcγRIIIa, FcγRIIIb) and C1q. Per detector (secondary antibody, FcγR or C1q), the median of all responders at 3 years post-SC was determined. The amount of detectors for which an individual had a response above this median was added up to obtain a score between 0 and 5. These scores were plotted to assess polyfunctionality. Polyfunctionality scores for the response at 6 months post-SC were based on the median of all responders at that time point. In addition, a polyfunctionality index was calculated by weighted addition of the percentage of individuals with each of the six possible polyfunctionality scores (0–5). The formula was as follows: $(0/5)^*\%n_0 + (1/5)^*\%n_1 + (2/5)^*\%n_2 + (3/5)^*\%n_3 + (4/5)^*\%n_4 + (5/5)^*\%n_5$. Where $\%n_x$ is the percentage of individuals with polyfunctionality score x. The result is a value between 0 and 100. This calculation is derived from the method of Larsen *et al.* [54].

Breadth scores were calculated with the same method, but by assessing the response to different HIV-1 strains used in our assays. Because we included 7 HIV-1 strains, the breadth score is a value between 0 and 7. Breadth scores were also determined separately for the response at 6 months post-SC. The breadth index was calculated with the following formula: $(0/7)^*\%n_0 + (1/7)^*\%n_1 + (2/7)^*\%n_2 + (3/7)^*\%n_3 + (4/7)^*\%n_4 + (5/7)^*\%n_5 + (6/7)^*\%n_6 + (7/7)^*\%n_7$.

Where %$n_x$ is the percentage of individuals with breadth score x. The result is again a value between 0 and 100.

To compare epitope specificity between different detectors, data normalization was required because of the large differences in quantity between the different Fc features. We used the z-score function in Matlab R2022b to mean center and variance scale the data across the six epitope scaffolds and gp120 and SOSIP proteins (8 antigens in total). This way, we conserved the relative differences between the different epitopes but made the data comparable between detectors. The resulting z-scores were plotted on radar charts in Excel (Microsoft Office 2016).

## Statistical analysis

Statistical tests and data visualization were performed using Graphpad Prism 9. Wilcoxon matched-pairs signed rank tests were used when comparing two paired groups. Mann-Whitney U tests were used when comparing two unpaired groups. A Kruskall-Wallis test followed by a Dunn's multiple comparison test was used when comparing more than two unpaired groups. Spearman regression was used for correlation analysis and the resulting P-values were corrected for multiple testing using the Bonferroni-Holm test in Matlab.

For multivariate analysis, we used partial least squares regression analysis (PLSR). First, all MFI and neutralization data was log transformed. Then, all data was transformed using the z-score function in Matlab. Feature selection was used to pre-select antibody Fc features before proceeding with further analysis. Elastic net feature selection was performed in Matlab using the Statistics and Machine Learning Toolbox with 100 iterations and 17-fold cross-validation. Different values of the Elastic-Net hyperparameter α were tested. The independent variable was either time between SC and AIDS in years or geometric mean neutralization titer.

PLSR was performed using Matlab and PLS Toolkit (Eigenvector). The log-transformed and z-scored features selected by the elastic net feature selection were used as x-data and the y-data were the same independent variables as used for feature selection, described above. Cross-validation was set at 12-fold, 18-fold or 20-fold depending on the amount of samples included. Scores, loadings and variable importance in projection (VIP) scores were used to assess the outcome of the model. As a negative control, the variable time to AIDS was scrambled for a Probability of Model Insignificance vs. Permuted Samples analysis in PLS Toolkit and showed that the predictions for our model were significantly different from permuted models ($P<0.001$ by Wilcoxon test).

## Supporting information

**S1 Table. The number of participants and data related to disease progression for participants subdivided in four groups based on time between seroconversion and AIDS.** Abbreviations: AIDS: acquired immunodeficiency syndrome; n: number; SC = seroconversion; cp/mL = viral copies per milliliter.
(DOCX)

**S1 Fig. Characterization of gp120 and SOSIP antigens generated for this study.** (A) Coomassie staining on BN PAGE gels loaded with protein antigens included in this study and control reagents. A marker (M) was included for a coarse estimation of size, but with preference of comparing the antigens with NIH-AIDS reagents; several monomer gp120 and gp140 control antigens were included. (B) Negative stain electron microscopy analysis of six SOSIP antigens. Micrographs (top) allow estimation of the degree of aggregation, while 2D classification (bottom) was used to estimate the amount of native-like trimer, malformed trimer and monomer/

dimer. (C) Heatmap of Log10 transformed area under the curve of median fluorescence intensity values obtained by Luminex assay using the SOSIP and gp120 antigens indicated on the x-axis on beads and assessing binding of the monoclonal antibodies indicated on the y-axis. Epitope clusters recognized by the mAbs are indicated. The plot is subdivided in two categories: broadly neutralizing antibodies (bNAbs) and non-neutralizing antibodies (non-NAbs). All combinations were tested, a cross indicates that no binding was detected. Abbreviations: MPER: membrane-proximal external region.

(TIF)

**S2 Fig. Levels of antibody features and responder rates over time.** (A) 92BR020-specific antibodies in serum targeting SOSIP and gp120 antigens of IgA and IgM type and interacting with FcγRI, FcγRIIa-R (low affinity allelic variant R131), FcγRIIb, FcγRIIIa-F (low affinity allelic variant F158) and FcγRIIIb. Antibody responses are compared between the time point 6 months (6 m) after seroconversion (SC) and 3 years (3 y) after SC using a Wilcoxon matched-pairs signed rank test. Only individuals with samples at both time points (n = 166) are included in this subfigure. Data is expressed as median fluorescence intensity (MFI) measured by Luminex assay. (B) Responder rates in percentage are indicated by the heatmap colors. On the left, in serum of 166 participants at 6 months post-SC, compared between four groups based on time between SC and acquired immunodeficiency syndrome (AIDS) diagnosis (less than 3 years (n = 20), between 3 and 7 years (n = 59), between 7 and 11 years (n = 21) and more than 11 years (n = 15)). On the right, in serum of 382 participants at 3 years post-SC, split based on disease progression in four categories (less than 3 years (n = 28), between 3 and 7 years (n = 147), between 7 and 11 years (n = 66) and more than 11 years (n = 71)). Responses are compared for 15 different 92BR020 SOSIP- and gp120-specific antibody features. ** = P <0.01, *** = P<0.001, **** = P<0.0001.

(TIF)

**S3 Fig. Additional analyses to assess associations between antibody types and antibody functionality and the effect of genetic signatures and allelic variation in FcγRs.** (A) Spearman correlations between the four subtypes of 92BR020 gp120-specific IgG and interaction of 92BR020 gp120-specific antibodies with all FcγRs and C1q for the full cohort (n = 382) at 3 years post-seroconversion (SC). (B) Spearman correlations between time between SC and acquired immunodeficiency syndrome (AIDS) diagnosis and 92BR020 gp120-specific antibodies interacting with FcγRIIa, FcγRIIIa and C1q. Results are shown for the full cohort with known time to AIDS (n = 312), individuals confirmed to not have the protective CCR5 deletion and with known time to AIDS (n = 231), individuals confirmed to not have the HLA-B57 gene and with known time to AIDS (n = 271) and individuals confirmed to not have the HLA-B27 gene and with known time to AIDS (n = 214). (C) Spearman correlations of time between SC and AIDS with 92BR020 gp120- and SOSIP-specific antibodies interacting with the two allelic variants of FcγRIIa and FcγRIIIa for the full cohort with known time to AIDS (n = 312). FcγRIIa-H and FcγRIIIa-V are the high affinity variants (FcγRIIa-H131 and FcγRIIIa-V158) while FcγRIIa-R and FcγRIIIa-F are the low affinity variants (FcγRIIa-R131 and FcγRIIIa-F158). (D) The color indicates the rho Spearman correlation coefficient and significant results are indicated by asterisks. The results were corrected for multiple comparisons using the Bonferroni-Holm method. * = P<0.05, ** = P <0.01, *** = P<0.001, **** = P<0.0001.

(TIF)

**S4 Fig. Polyfunctionality of Fc features at 6 months post-serocoversion.** (A) Polyfunctionality was assessed by calculation of a polyfunctionality score and index at 6 months post-

seroconversion (SC). We only assessed activating FcγR (FcγRI, FcγRIIa, FcγRIIIa, FcγRIIIb) and C1q interaction, and for each Fc feature we determined for each participant if there was a response that was higher than the median response of all responders in the cohort. The resulting scores (amount of features above the responder median, between 0 and 5) were plotted for the four groups based on time between SC and acquired immunodeficiency syndrome (AIDS) diagnosis (less than 3 years (n = 20), between 3 and 7 years (n = 59), between 7 and 11 years (n = 21) and more than 11 years (n = 15)) with one plot for gp120-specific responses and one plot for SOSIP-specific responses. In addition, a polyfunctionality index was calculated for each group as a quantitative measure of polyfunctionality. This is a value between 0 and 100 calculated by weighted addition of the percentage of individuals in each category based on the number of features, derived from Larsen *et al.*[54], see methods). This index is shown above each bar.
(TIF)

**S5 Fig. IgG response against seven HIV-1 strains compared to neutralizing antibody responses.** (A) On the left, responder rates in percent for IgG binding of seven HIV-1 strains. Shown for gp120-specific and SOSIP-specific responses, for 166 individuals at 6 months post-seroconversion (SC) and 382 individuals at 3 years post-SC. On the right, percentages of participants with detectable neutralizing antibodies in 382 individuals at 3 years post-SC. Neutralization assays were not performed for strain BG505. Responders are shown in blue and non-responders in purple. (B) On the left, distribution of participants with no neutralizing capacity against any tested strain and with neutralizing capacity against one, two, three, four, five or six strains. On the right, similar plots showing the distribution of participants with detectable gp120-specific IgG or SOSIP-specific IgG. IgG levels were tested against seven strains. (C) Spearman correlations between SOSIP-specific IgG and gp120-specific IgG, both expressed as median fluorescence intensity (MFI), with the Spearman rho correlation coefficients and P-values indicated on each graph. (D) Spearman correlations between half-maximal infective dilution ($ID_{50}$) pseudovirus neutralization titer and SOSIP- or gp120-specific IgG expressed as MFI. Spearman rho correlation coefficients and P-values are indicated on each graph.
(TIF)

**S6 Fig. Breadth of gp120- and SOSIP-specific antibody interaction with FcγRs and C1q.** (A) Breadth of SOSIP-specific antibody interaction with FcγRI, FcγRIIa, FcγRIIb, FcγRIIIa, FcγRIIIb and C1q at 3 years post-seroconversion (SC) is compared between four categories based on time between SC and acquired immunodeficiency syndrome (AIDS) diagnosis: less than 3 years (n = 28), between 3 and 7 years (n = 147), between 7 and 11 years (n = 66) and more than 11 years (n = 71). (B) Breadth of gp120-specific antibody interaction with FcγRI, FcγRIIa, FcγRIIb, FcγRIIIa, FcγRIIIb and C1q at 6 months post-SC is compared between four categories based on time between SC and AIDS: less than 3 years (n = 20), between 3 and 7 years (n = 59), between 7 and 11 years (n = 21) and more than 11 years (n = 15). (C) Breadth of SOSIP-specific antibody interaction with FcγRI, FcγRIIa, FcγRIIb, FcγRIIIa, FcγRIIIb and C1q at 6 months post-SC is compared between the same four categories based on time between SC and AIDS. Bar charts show the percentage of individuals within each group with antibody interaction with the plotted FcγR or C1q above the responder cut-off for the color-coded amount of strains. Above each bar the breadth index for that group is shown as a quantitative measure of breadth. This is a value between 0 and 100 calculated with a similar calculation as for polyfunctionality in Fig 2, by weighted addition of the percentage of individuals in each category based on the number of features, derived from Larsen *et al.*[54].
(TIF)

**S7 Fig. Responder rates for all measured antibody features specific for SOSIP, gp120 and all constructs presenting HIV-1 Env epitope clusters.** Responder rates in percentage are indicated by the heatmap colors. On the left, in serum of 166 participants at 6 months post-seroconversion (SC), compared between four groups based on time between SC and acquired immunodeficiency syndrome (AIDS) diagnosis (less than 3 years (n = 20), between 3 and 7 years (n = 59), between 7 and 11 years (n = 21) and more than 11 years (n = 15)). On the right, in serum of 382 participants at 3 years post-SC, split based on disease progression in four categories (less than 3 years (n = 28), between 3 and 7 years (n = 147), between 7 and 11 years (n = 66) and more than 11 years (n = 71)). (A) For 15 different 92BR020 SOSIP- and gp120-specific antibody features and (B) for 10 different antibody features specific for six constructs presenting HIV-1 Env epitope clusters. Abbreviations: V1/V1: variable regions 1 and 2; V3: variable region 3; CD4bd: CD4 binding site; MPER: membrane-proximal external region; gp120-CD4: a covalently linked fusion of gp120 and CD4.
(TIF)

**S8 Fig. Additional figures for the partial least squares regression analysis for the association between antibody Fc features and disease progression and correlation between ADCP and prominent features** (A) Outcomes of six rounds of elastic net feature selection with different values for alpha and 17-fold cross-validation as performed for the partial least squares regression analysis shown in Fig 5. The number of selected variables is plotted on the left y axis and the $R^2$ of calibration of the partial least squares regression analysis is shown on the right y axis. Alpha 0.8 was chosen, as indicated by the grey area. (B) Additional model information for the partial least squares regression analysis shown in Fig 5. Measured values of y are time between seroconversion (SC) and acquired immunodeficiency syndrome (AIDS) diagnosis in years as recorded in the cohort database while y predicted are the predictions of the model. (C) Spearman correlation analysis between antibody-dependent cellular phagocytosis (ADCP) of 92BR020 gp120-conjugated beads by THP-1 cells at 3 years post-SC and 92BR020 gp120-specific antibody interaction with FcγRIIa and (D) FcγRI. The ADCP assays were done on a subset of 140 participants with a similar distribution of time between SC and AIDS as the full cohort. r = rho Spearman correlation coefficient, Abbreviations: RMSEC: Root mean square error of calibration; RMSECV: Root mean square error of cross-validation, MFI: median fluorescence intensity.
(TIF)

**S9 Fig. Associations between antibody Fc features and broad HIV-1 neutralization activity.** (A) Spearman correlation analysis between the antibody features analyzed in this study at 3 years post-seroconversion (SC) and geometric mean (geomean) half-maximal infective dilution ($ID_{50}$) pseudovirus neutralization titers at 3 years post-SC (n = 382). Antibody features shown in this figure are for responses specific for gp120 and SOSIP proteins of subtype B strain 92BR020. The color indicates the rho Spearman correlation coefficient and significant results are indicated by asterisks. The results were corrected for multiple comparisons using the Bonferroni-Holm method. (B) Spearman correlation analysis between the antibody features analyzed in this study at 3 years post-SC and geomean $ID_{50}$ pseudovirus neutralization titers at 3 years post-SC (n = 382). Antibody features shown in this figure are for responses specific for gp120 and SOSIP proteins of subtype B strain JRCSF, V1/V2, V3 and gp41 epitope constructs were also based on the JRCSF sequence. Constructs comprising the CD4 binding site (CD4bs), the membrane-proximal external region (MPER) and a covalently linked fusion of gp120 and CD4 (gp120-CD4) were previously described (see methods). The color indicates the rho Spearman correlation coefficient and significant results are indicated by asterisks. The results were corrected for multiple comparisons using the Bonferroni-Holm method. (C) Partial least

squares regression analysis with geomean $ID_{50}$ pseudovirus neutralization titer as independent variable. Twenty-three Fc features were included in the model as selected by elastic net feature selection with alpha 0.5 and 20-fold cross-validation. The model comprised two latent variables (LVs). (D) Additional model information. (E) Loading and variable importance in projection (VIP) scores of all 23 included features, with variables with a VIP score higher than 1 shown in light blue (also indicated by the grey vertical line) and the remaining variables shown in dark blue. Abbreviations: V1/V1: variable regions 1 and 2; V3: variable region 3; CD4bd: CD4 binding site; MPER: membrane-proximal external region; RMSEC: Root mean square error of calibration; RMSECV: Root mean square error of cross-validation. * = P<0.05, ** = P <0.01, *** = P<0.001, **** = P<0.0001.
(TIF)

**S10 Fig. Coordination between Fc polyfunctionality and neutralization breadth and titer.** (A) Polyfunctionality was assessed by calculation of a polyfunctionality score at 3 years post-seroconversion (SC) (n = 382). We only assessed activating FcγR (FcγRI, FcγRIIa, FcγRIIIa, FcγRIIIb) and C1q interaction, for the gp120 and SOSIP protein of subtype B strain 92BR020. For each Fc feature we determined for each participant if there was a response that was higher than the median response of all responders in the cohort. The resulting scores (amount of features above the responder median, between 0 and 5) were plotted for seven groups based on neutralization breadth (categorized by the number of strains neutralized). (B) The same analysis was done for antibody Fc polyfunctionality at 6 months post-SC (n = 166). (C) A similar analysis for antibody Fc polyfunctionality at 3 years post-SC (n = 382) but with categories based on 92BR020 half-maximal infective dilution ($ID_{50}$) pseudovirus neutralization titer. For each Fc feature we determined for each participant if there was a response that was higher than the median response of all responders in the cohort. The resulting scores (amount of features above the responder median, between 0 and 5) were plotted for four groups based on neutralization titer. (D) The same analysis was done for antibody Fc polyfunctionality at 6 months post-SC (n = 166). In addition, a polyfunctionality index was calculated for each group as a quantitative measure of polyfunctionality. This is a value between 0 and 100 calculated by weighted addition of the percentage of individuals in each category based on the number of features, derived from Larsen et al.[54], see methods). This index is shown above each bar.
(TIF)

## Acknowledgments

We would like to express our sincere gratitude to the participants of the Amsterdam Cohort Studies. The Amsterdam Cohort Studies on HIV infection and AIDS, a collaboration between the Public Health Service Amsterdam, the Amsterdam UMC of the University of Amsterdam, Medical Center Jan van Goyen and the HIV Focus Center of the DC-Clinics, are part of the Netherlands HIV Monitoring Foundation and financially supported by the Center for Infectious Disease Control of the Netherlands National Institute for Public Health and the Environment. We thank Ronald Derking, Iván del Moral Sánchez and Kwinten Sliepen of the Amsterdam UMC, Kevin J. Selva, Pradhipa Ranamathan, Milla McLean, Timon Damelang, Samantha K. Davis and Sarah Collins of the University of Melbourne for sharing their expertise and scientific discussion. We are grateful to William R. Schief of the Scripps Research Institute and George K. Lewis from the University of Maryland School of Medicine for providing protein antigens.

## Author Contributions

**Conceptualization:** Marloes Grobben, Marit J. van Gils.

**Formal analysis:** Marloes Grobben.

**Funding acquisition:** Marit J. van Gils.

**Investigation:** Marloes Grobben, Angela I. Schriek, Liesbeth J.J. Levels, Jeffrey C. Umotoy, Khadija Tejjani, Ryan N. Lin, Gabriel Ozorowski.

**Methodology:** Marloes Grobben, Angela I. Schriek, Steven W. de Taeye, Amy W. Chung.

**Resources:** Margreet Bakker, Angela I. Schriek, Jeffrey C. Umotoy, Mariëlle J. van Breemen, Steven W. de Taeye, Neeltje A. Kootstra, Andrew B. Ward, Stephen J. Kent, P. Mark Hogarth, Bruce D. Wines, Rogier W. Sanders.

**Supervision:** Rogier W. Sanders, Amy W. Chung, Marit J. van Gils.

**Validation:** Marloes Grobben.

**Visualization:** Marloes Grobben.

**Writing – original draft:** Marloes Grobben.

**Writing – review & editing:** Marloes Grobben, Margreet Bakker, Khadija Tejjani, Mariëlle J. van Breemen, Ryan N. Lin, Gabriel Ozorowski, Neeltje A. Kootstra, Andrew B. Ward, Stephen J. Kent, P. Mark Hogarth, Bruce D. Wines, Rogier W. Sanders, Amy W. Chung, Marit J. van Gils.

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
