## [Decision Letter · Decision Letter 0]

29 Apr 2024

Dear Ms. Grobben,

Thank you very much for submitting your manuscript "Polyfunctionality and breadth of HIV-1 antibodies are associated with delayed disease progression" for consideration at PLOS Pathogens. As with all papers reviewed by the journal, your manuscript was reviewed by members of the editorial board and by 2 independent reviewers. The reviewers appreciated the attention to an important topic. Based on the reviews, we are likely to accept this manuscript for publication, providing that you modify the manuscript according to the review recommendations.

Sincerely,

Penny L. Moore

Academic Editor

PLOS Pathogens

Susan Ross

Section Editor

PLOS Pathogens

Michael Malim

Editor-in-Chief

PLOS Pathogens

orcid.org/0000-0002-7699-2064

Reviewer Comments (if any, and for reference):

Reviewer's Responses to Questions

**Part I - Summary**

Reviewer #1: In this study, Grobben et al. conducted a considerable analysis to investigate the role of different antibody features on HIV disease progression using samples from a large cohort of untreated people living with HIV. Numerous univariate associations were found between various antibody features, including breadth and polyfunctionality of responses, and delayed disease progression or neutralization breadth/potency. Multivariate analysis indicated FcgRIIA may play an important role. The study is dense with numerous similar analyses which sometimes made it difficult to follow (for instance, Figure 2, S6 and S8 have the same/very similar figure legends) but, overall, the study is an important addition to understanding the many factors that likely contribute to delayed HIV progression and provides further support that polyfunctional antibodies will be important in future HIV vaccine and cure strategies.

Reviewer #2: This study uses a data dense methodology to unpick Fc binding and isotype antibody signals associated with the development of HIV neutralization breadth and slowed disease progression to AIDS. While i believe this manuscript to offer unique insights, I find several of the findings overstated and based on weak correlations. I also suggest that some functional data would greatly strengthen the publication and given the expertise of the authors included should not be onerous to perform, especially given the emergence of FcyR2A as a major signature. Several minor suggestions and discussion points are included.

**Part II – Major Issues: Key Experiments Required for Acceptance**

Reviewer #1: (No Response)

Reviewer #2: 1. I recommend that functional data be included to at least support the breadth associations for FcyRIa and FcyRIIa(ADCP) with slowed disease progression. While interesting that FcyRIIa is a major signature, it is not the only determinate of ADCP and as such to make the claim that ADCP is associated with protection this should be tested, if only in a subset.

**Part III – Minor Issues: Editorial and Data Presentation Modifications**

Reviewer #1: Aspects requiring clarification:

- The results section (line 120) suggests at least some of this cohort remained ART-naïve up to 11+ years but the method section (study population) is unclear if this cohort remained ART-naïve after the study period ended, stating that participants were ART-naïve until 3 years post-seroconversion (line 526). It seems apparent that these participants remained ART-naïve until AIDS diagnosis or for 11+ years but a clarifying statement would alleviate the concern that ART use at some point during the years post seroconversion.

- It is interesting that the authors decided to undertake the analyses on the full set of participants, including those with genetic factors known to influence disease progression, such as HLA-B57/HLA-B27 and deltaCCR5 mutations. Why were these not removed to avoid these known potential confounders and, perhaps, strengthen the associations in the remaining subset of participants? What was the distribution of these participants in time to AIDS? And did they exhibit any different antibody features to those without known genetic factors?

- The majority of binding assays were done using 92BR020 gp120 and SOSIP, but the V1V2, V3 and gp41 proteins used JRCSF. It is clear from the breadth analyses that there are fewer neutralizers against JRCSF than 92BR020, and correlations between neutralization and JRCSF SOSIP binding, as well as between JRCSF SOSIP and gp120 binding, is lower than 92BR020 neutralization and binding, respectively. How similar are the 92BR020 and JRCSF V1V2, V3 and gp41 sequences? It seems possible that lower recognition of JRCSF V1V2 specifically could have contributed to the lack of correlation with progression to AIDS.

- Similarly, the gp120-CD4 construct, assuming it is FLSC as suggested in figure S12, consists of BaL gp120 which is lab adapted and typically has very high binding (evident in S10). Could this have contributed to decreasing the correlation with disease progression?

- It is not clear if the comprehensive profiling of antibody features (lines 315-325) is conducted using antibody feature data at 6 months post sero-conversion and 3 years post-seroconversion, or just 3 year post sero-conversion. If both, it does not appear there was any factoring for the dependency/shared effect of that subset of samples? Does this analysis change if using antibody feature data at 6 months or 3 years post-seroconversion separately?

Minor issues:

- Line 191: HLA-B57 (sp.)

- There were no obvious references for the MPER and gp120-CD4 constructs. Line 303 suggests they are in the methods but do not appear to be there.

- The gp120-CD4 construct is labelled FLSC in Figure S12. Change for consistency.

- Figures S7C and 7D are switched in the text (lines 233-235). In Figure S7, C refers to correlation between neutralization and SOSIP binding while D refers to SOSIP and gp120 binding.

- Lines 282-283: the increased gp120-CD4 binding by antibodies that bound C1q at < 7 years, otherwise binding seems comparable for > 7 years?

- Line 682: Bonferroni-Holm test (sp.)

- Supplementary table 1 part 1 and 2 (samples at 3 years and 6 months post-seroconversion available): “time since SC” states years but is actually months. Same in Part 2.

Reviewer #2: 1. Abstract: The authors state that “the capacity to interact with all Fc γ receptors (FcγRs) and C1q, and in particular with FcγRIIa, correlated positively with delayed disease progression”. All Fc receptors would also imply FcγRIIb which is predominately inhibitory?

2. Introduction: Rephrase the sentence in line 66: “The effectivity of antibody effector functions is determined by a range of factors.” The term effectivity is a strange one which is used repeatedly.

3. In line 85 you can also add SARS-CoV-2 – interesting new preprint from Florian Krammer 10.1101/2024.02.28.582613

4. The paragraph from line 100 feels repetitive, I think the rationale is clear without multiple repeating lines.

5. It should be made clear that the authors are not looking at Fc effector functions – They are measuring Fc receptor and complement binding as proxies for the actual assays. Albeit they typically good proxies, much of what is known about Fc receptor binding is not always a perfect correlation with the functional output. Further, neutralization is an assay measure of function and the Fc array a measure of binding.

6. It is not clear what is meant by polyfunctionality in this study given no functions were tested, perhaps another term is more correct?

7. I am not clear as to why such detailed characterisation is focused on 92BR020 SOSIP. Appreciating that this in itself is a lot of work, if it’s because it has not been previously characterised then this should be clarified in the text.

8. Do the non-responders in the SOSIP groups for Figure 1 correlate with neutralization against the matched virus?

9. Sup Fig 2: it is not clear how the authors decided on which features to show in the main text and which to show in the supp?

10. Sup Fig 2B – on the figure it would be helpful to have a key for “time to AIDS” also from Fig 1A in the main text seems like everyone has IgG3 levels but this figure doesn’t show that?

11. Line 177: double full stop

12. The correlations in 1B, although significant are extremely weak and do not show a particular meaningful correlation – in the results the authors state that these are strong which should be corrected. As such it is difficult to suggest that these features are associated with disease progression

13. While the polyfunctionality score is clear , it is not clear what the polyfunctionality index shows that is more informative than the score.

14. Authors state that 25% of these participants were previously determined to have a bNAb response (defined as neutralizing >4 out of 6 strains with an ID50>10047) . This should also make reference to the Simek paper that defined this downselected panel.

15. In general there are too many supplementary figures and I found this made reading the study overwhelming, I suggest decreasing non-essential data

16. It is stated that the neutralization titer correlated better with SOSIP than gp120, as expected but the correlation with many of the timers is weak. Given that several of the worst correlating trimers were also those with aggregation, is this data against these trimers reliable and representative of trimer binding?

17. For the constructs representative of different epitopes, the strains from which they were derived should be stated in the main text. The association with slowed disease progression should also be tempered here with the statement that only specific strains were used.

18. Again the correlations shown in Figure 4a are significant but weak, as such conclusions should be tempered.

19. Are the differences shown in Figure 4B significant? Again these seem very slight.

20. The authors state “Breadth was not identified as important, whereas polyfunctionality did assume a clear role in the model.” Can this be clarified?

21. Discussion: While interesting that FcyRIIa is a major signature, it is not the only determinate of ADCP and as such to make the claim that ADCP is associated with protection this should be tested, if only in a subset. Further the authors go on to mention that FcyRIIa binding could be improved in bNAbs but perhaps more mention should also be made to FcyIIB interaction which would also have to be considered to balance out effects

22. Is the association seen with IgG1 just a titer effect? Ie it is just associating so strongly because there is more in the plasma. This may be relevant given that the author later postulates that they may have missed V1/V2 associations etc because of titer.

23. The authors state that their study is unique because they are unpicking the mechanisms of slowed progression but they go on to say that nothing can show causality – perhaps in the introduction the authors show strengthen the novelty of their study.

24. Why were the associations for IgA were limited in your analysis?

25. “We found that antibodies with functional Fc features, especially when interacting with C1q more often recognize CD4-i epitopes and the V3 region” – can the authors postulate as to why?

26. More mention should also be made of the fact that the strains of v3 , v2 etc are limited and may contribute to skewing of the patterns seen.

27. The observation that Fc is associated with neutralization breadth but also slowed disease progression is interesting but in the discussion, the differences in signatures should be more clearly contrasred. The association of any SOSIP binding titers is expected but it is interesting that FcyR2a v3 is a signal for breadth when v3 binding mAbs are often a hinderance for breadth.

28. Were the bNAbers matched in viral load for non-bNAbers, as a major driver of breadth this is important to mention.

29. In light of the data for the association with breadth the authors may consider changing the title to encompass this finding

30. The final conclusion “In conclusion, our study identifies correlates for prolonged control of HIV-1 disease progression” should be tempered given weak associations and rather focused on the second half of that statement “provides a specific biophysical profile of antibody features attractive for preventative and therapeutic strategies against HIV-1”.

PLOS authors have the option to publish the peer review history of their article (what does this mean?). If published, this will include your full peer review and any attached files.

Reviewer #1: No

Reviewer #2: **Yes: **Simone I. Richardson

Figure Files:

Data Requirements:

Reproducibility:

References:

---

## [Editor Report · Decision Letter 1]

9 Nov 2024

Dear Ms. Grobben,

We are pleased to inform you that your manuscript 'Polyfunctionality and breadth of HIV-1 antibodies are associated with delayed disease progression' has been provisionally accepted for publication in PLOS Pathogens.

Best regards,

Penny L. Moore

Academic Editor

PLOS Pathogens

Susan Ross

Section Editor

PLOS Pathogens

Michael Malim

Editor-in-Chief

PLOS Pathogens

orcid.org/0000-0002-7699-2064
---

## [Editor Report · Acceptance letter]

22 Nov 2024

Dear Ms. Grobben,

We are delighted to inform you that your manuscript, "Polyfunctionality and breadth of HIV-1 antibodies are associated with delayed disease progression," has been formally accepted for publication in PLOS Pathogens.

Best regards,

Michael Malim

Editor-in-Chief

PLOS Pathogens

orcid.org/0000-0002-7699-2064